# Updating the Methodology of Identifying Maize Hybrids Resistant to Ear Rot Pathogens and Their Toxins—Artificial Inoculation Tests for Kernel Resistance to *Fusarium graminearum*, *F. verticillioides*, and *Aspergillus flavus*

**DOI:** 10.3390/jof8030293

**Published:** 2022-03-11

**Authors:** Akos Mesterhazy, Denes Szieberth, Eva Tóth Toldine, Zoltan Nagy, Balázs Szabó, Beata Herczig, Istvan Bors, Beata Tóth

**Affiliations:** 1Cereal Research Non-Profit Ltd., P.O. Box 391, 6701 Szeged, Hungary; zsigi2011@gmail.com (E.T.T.); zoltan.nagy@gabonakutato.hu (Z.N.); balazs.szabo@gabonakutato.hu (B.S.); beata.toth@gabonakutato.hu (B.T.); 2Hungarian Maize Club, Kazinczy Street 12, 8152 Fejér, Hungary; magyarkukoricaklub@me.com; 3Bonafarm-Babolna Feed Ltd., Laboratory Branch, 2942 Nagyigmand, Hungary; Bea.Herczig@btakarmany.bonafarm.hu (B.H.); Istvan.Bors@btakarmany.bonafarm.hu (I.B.)

**Keywords:** maize, *Fusarium graminearum*, *F. verticillioides*, *Aspergillus flavus*, resistance to ear rots, reduction of toxins, preharvest toxins, estimation of toxin risk, prevention of toxin contamination

## Abstract

Resistance to toxigenic fungi and their toxins in maize is a highly important research topic, as mean global losses are estimated at about 10% of the yield. Resistance and toxin data of the hybrids are mostly not given, so farmers are not informed about the food safety risks of their grown hybrids. According to the findings aflatoxin regularly occurs at preharvest in Hungary and possibly other countries in the region can be jeopardized. We tested, with an improved methodology (two isolates, three pathogens, and a toxin control), 18 commercial hybrids (2017–2020) for kernel resistance (%), and for toxin contamination separately by two–two isolates of *F. graminearum*, *F. verticillioides* (mg/kg), and *A. flavus* (μg/kg). The preharvest toxin contamination was measured in the controls. Highly significant kernel resistance and toxin content differences were identified between hybrids to the different fungi. Extreme high toxin production was found for each toxic species. Only about 10–15% of the hybrids showed higher resistance to the fungal species tested and lower contamination level of their toxins. The lacking correlations between resistance to different fungi and toxins suggest that resistance to different fungi and response to toxin contamination inherits independently, so a toxin analysis is necessary. For safety risk estimation, separated artificial and natural kernel infection and toxin data are needed against all pathogens. Higher resistance to *A. flavus* and *F. verticillioides* stabilizes or improves feed safety in hot and dry summers, balancing the harmful effect of climate changes. Resistance and toxin tests during variety registration is an utmost necessity. The exclusion of susceptible or highly susceptible hybrids from commercial production results in reduced toxin contamination.

## 1. Introduction

Maize was globally the most important cereal produced in 2019 and 2020 [1]. Maize is commonly affected by three main toxigenic fungi: *F. graminearum*, producer of deoxynivalenol and zearalenone, *F. verticillioides*, synthetizing more than hundred fumonisins (where FUM B_1_ and B_2_ have the highest importance), and *A. flavus* producing aflatoxins, the most important of which is AFB1. During warm and humid seasons, *F. graminearum* is more commonly found in maize; warmer and rather dry conditions are favorable to *F. verticillioides*; and *A. flavus* is common in the warmest corn areas [2,3,4]. The ecological needs of these fungi are different; therefore, they can occur alone or combined. In regions with crossroads of different climatic influences, such as Hungary, all three may be present with their toxins in one year. The disease resistance and toxin resistance of various hybrids might be different [5]. Reliable information on maize resistance to toxigenic fungi has seldom been provided for commercial hybrids [6]; therefore, an evaluation of the production risks of hybrids for food safety is a global but mostly neglected task. In the commercial hybrids, no inbred lines can be tested. Therefore, genetic work is not possible. There is no reason to test inbreds in this respect, as none are used in commercial production. To influence commercial production, information about the ear rot resistance and food safety risks of the hybrids is needed to choose the less susceptible ones. An improved understanding of the ear rot resistance, the toxin relations, the artificial and natural infection comparisons, and toxin data with respect to different toxigenic fungi may help to provide farmers more reliable information on the risk of hybrids. These data are similarly important for farmers, breeders, animal husbandry, and the ethanol industry. 

The economic losses due to toxigenic fungi are substantial [7], with corn loss estimated at ranging from 52 million to 1.68 billion USD annually in the U.S. as climate change further progresses, and one review [8] lists many other literature sources on the global toxin situation. In Hungary, the 2014 mycotoxin contamination caused 330 million USD in damage to corn production and animal husbandry [5]. At least 10% of the global harvested grains is toxin-contaminated above limit values. Adopting this number for maize as well (about 100 MMT) is a problem that requires a more rapid treatment than traditional plant breeding can suggest [8]. 

There are at least 19 *Fusarium* species from Europe [9], and they produce a substantial number of mycotoxins. Two of them have global significance: *F. graminearum*, which produces deoxynivalenol (DON) and zearalenone (ZEA), as well as *F. verticillioides*, which synthesizes the members of the fumonisin group [10]. In Hungary, similar *Fusarium* species occur [11] with similar significance. *Aspergillus* spp. have occurred in Hungary decades before, but the high rate of *Penicillium* spp. proved that samples originated from stored samples [12]. However, the preharvest origin of the aflatoxin could not be verified until now. *A. flavus* has been accepted as a preharvest aflatoxin producer in the U.S. [10] (postharvest contamination also remains significant). Knowing this, we started testing for Aspergillus resistance, as we supposed that Hungary would eventually face the same problem when summer temperatures increase. When preharvest AFB1 is above 20 μg/kg, preventive methods with a higher resistance, supported by agronomy and other means, are crucial [13,14]. Toxin data from SGS Company [15] showed that DON, fumonisins, and AFB1 have been regular components of corn grains every year since 2012, with large local differences. As the data (except the low rates) originated from mixtures from different fields and hybrids [16], the data could not be connected to individual hybrids. In the U.S., a significant part of AFB1 is of field origin [10]. Therefore, the inclusion of *A. flavus* in the resistance test was a logical step to prepare for the preharvest identification of *A. flavus* resistance and aflatoxin screening in Hungary. 

The global climate models forecasting significant temperature increases [17,18] reported that AFB1 contamination would increase with a 2 °C increase in South Germany throughout the full Carpathian Basin. Other authors reached similar conclusions [19,20]. 

The AFB1 production of *A. flavus* in maize is significantly influenced by the false positive isolates with full cluster for AFB1 synthesis but without actual toxin production ability. In Kenya two maize populations were screened for toxic rate of *A. flavus* isolates: 71% and 62% of the isolates produced aflatoxin on coconut agar [21]. In India 63% and 52% of the *A. flavus* isolates were AFB1 producer in maize grains [22]. In our lab, 42 *A. flavus* isolates produced AFB1 in rice medium with the presence of the whole aflatoxin gene cluster, but only eight produced AFB1 on maize ears. Thirty-four can be classified as false positive (Toth 2019, unpublished). To differentiate the false positive isolates from the positive a quintuplex RT-PCR procedure was suggested that could differentiate the false positive isolates from the real toxin producer ones [23].

Magan et al. [24] analyzed DON for wheat, fumonisins, and AFB1 in maize and found that increasing warmth and dryer weather will favor fumonisins and aflatoxins in maize, and warm temperatures with excessive rain will increase DON in wheat. This also applies to DON contamination in maize [15,25].

Areas such as Mediterranean Region are today at high risk for aflatoxins and fumonisins because the temperature is now optimum or higher than optimum for fungal infection and toxin production. This is not yet the case in Middle and Western Europe, but a higher toxin pressure was forecasted [24]. A systemic infection at high temperatures by *F. verticillioides* [26] increases additional risk for FER infections. 

The probability of infection mediated by rachilla is possible with all three pathogens [27,28,29,30]. The conclusion is that regions below the optimum for diseases or toxins will be more exposed in warmer seasons. The severely hit Mediterranean regions will remain in this category [18] and need help. We know that the moisture content of the cob is significantly higher, which allows the fungus to grow on the surface of the cob and infect the grain and germs, which cannot be detected from the surface [31]. Epidemiologically, this is a problem. As in earlier hybrids, the drydown of the ear and cob is closer, and their use can decrease this additional risk, as seen in Hungary in 2014. 

The literature distinguishes two resistance types for ear rots [32]: kernel resistance and silk channel resistance. Kernel resistance (KR) is tested by growing fungus on toothpicks or dipping toothpicks or steel needles in a suspension and then inoculating in the middle of the ear. Silk channel resistance (SR) is tested by injecting a given amount of suspension into the silk channel, and the infection will be mediated by the silks to the grains. Normally, there are significant correlations between the responses of the genotypes and the two types [25,33,34,35,36,37]. In the Hungarian dryer and warmer conditions, kernel ear rot severity is threefold higher than the silk channel data, providing an improved differentiation of the 44 genotypes [25]. KR provided a much clearer differentiation of the genotypes. Therefore, kernel resistance was chosen for standard tests, even though it seems to be a well-supported fact that most ear inoculations are silk mediated [10]. 

Resistance relationships vary between toxigenic species. Most breeding programs have concentrated on a single toxigenic species, mostly *F. verticillioides*, and much less on *F. graminearum*. Several authors have reported significant correlations between resistance to *F. graminearum* and *F. verticillioides* [38,39,40,41]. There are examples of correlated results between *F. verticillioides* and *A. flavus* resistance or *F. graminearum* and *F. verticillioides*, but no data exist for the three pathogens in a single research program. Löffler et al. [40] found genotype/environment interactions between GER and FER, but this was never extended to all tested genotypes. The conclusion of Rose et al. [13] on the relation between *F. verticillioides* and *A. flavus* was the same. In recently tested hybrids [5,25], 10–15% showed similar resistance to the three pathogens, and 5–10% showed high susceptibility to all; and 75–80% of the hybrids showed a highly variable resistance to the different pathogens. In several hybrids, a strong toxin overproduction has been found, e.g., a much higher toxin contamination than would follow from visual ear rot severity [5,25]. There are several papers describing medium to close relationships between resistance to *F. verticillioides* and *A. flavus*, and some FER-resistant maize inbred lines can be a source of *A. flavus* and AFB1 resistance [41,42]. Other authors have identified AFB1 and fumonisin resistance (the resistance to disease and resistance to toxin accumulation were treated as synonyms) in two new inbred lines in a mapping population by indicating the lack of general agreement [43,44]. Therefore, there is likely no general agreement on the resistance to these pathogens. The conclusion is that resistance to different toxigenic species should be tested separately. A general ear rot resistance does not exist.

Significant correlations were found between the severity of visual symptoms and toxin contamination; generally, they were of medium closeness or were somewhat stronger [45,46,47,48]. Bolduan et al. [49] calculated an unusually close correlation between ear rot and DON (r = 0.94) with respect to Fusarium ear rot (FER); the conclusion was that toxin analysis may not be necessary. In other cases, much lower correlations were found [3]. Correlation-breaking genotypes were also identified, indicating that a general agreement in maize would not be the case. Löffler et al. [35] analyzed the relationship between toxin contamination and visual ear rot infection severity in maize inbreds to *F. graminearum* and *F. verticillioides*. They found that separate divergent behavior is not an exception. For this reason, parallel testing of resistance to both pathogens were suggested. A strong agreement was found between *F. graminearum* and *F. culmorum* [5,50,51]. Other data also support this view [5,25,50]. 

The reduction in disease symptoms due to resistance has seldom been discussed, and even less so for toxins. Focker et al. [52] reported a reduction in AFB1 of between 62% and 82%. Whether we can reduce toxin contamination below the official limit is another question. We think that the reduction rate is not sufficient, and that the contamination level should be below the EU limits. 

In terms of the methodology, except in the pathogenicity tests performed by Mesterházy et al. [4], all authors have worked with one isolate or a mixture of isolates, which has remained unchanged until now [25]. Based on initial experimental results on wheat, maize hybrids inoculated with different independent isolates have been evaluated [50,51,53]. In the first experiment [50], 14 isolates (10 *F. graminearum* and 4 *F. culmorum*) were assessed on 10 hybrids. Of the 55 possible correlations in the hybrid reactions, 22 were not significant, and Hybrid 2 did not have any significant relationships with the reactions to other hybrids. Hybrid 10 showed significant correlations with all other hybrids, except Hybrid 2. In another test, four isolates were applied independently without mixing [54]. The severity of the disease and the toxin contamination were highly variable. The differences between years were also significant, which is no surprise. Other results [55] also support the view that more isolates are necessary to obtain a more reliable picture of resistance and toxin relationships. 

In risk analysis, different philosophies can be applied. Traditional natural infection indicates that highly infected genotypes can be excluded; however, in Hungary, such severity occurs once or twice in a decade. As different years might show differing fungal species compositions, screening for complex resistance on this basis is impossible [11]. For natural toxin contamination, we used EU limit values directly [56,57,58,59]. 

The conclusion is that a high rate of commercial hybrids is highly susceptible to one or more major toxigenic species. However, a smaller but significant part of the hybrids performs well with *Fusarium* spp. [5,25]. This is clear proof that commercial hybrid programs do not pay enough attention to the screening of hybrids for breeding resistance to toxigenic species and to inhibit toxin contamination. As the aflatoxin response could not be measured earlier [5], we aimed to include it into the testing program. Strategically, the most important step is the screening of the hybrids, as preference of more resistant hybrids can lead to a rapid improvement in food safety. The extent to which this is possible with three toxigenic species and three largely differing toxins is the question we seek to answer. Screening of the inbreds will be necessary, but their combining abilities to toxigenic species and yield are probably different. Therefore, their complex value can be decided better in their hybrids.

Our objectives were as follows: 

Analyzing the relationships between symptom severity and toxin contamination and to analyze the behavior of hybrids under different ecological conditions to further understand natural infection and toxin contamination; determining how the risk of a given hybrid can be evaluated, which traits should be considered, and how they should be weighted in the analysis and evaluating the influence of the growing resistance level to better tolerate the higher amplitudes of the changing climate, which causes mostly higher temperatures and draught and is favorable to fumonisin and aflatoxin production.

## 2. Materials and Methods

### 2.1. Plant Material and Experimental Design

Maize hybrids were chosen from the registered hybrids in Hungary, from Corteva Agriscience (Pioneer, Dow AgroSciences Hungary Kft. 2040 Budaörs, Neumann János str. 1., Hungary), Bayer (DeKalb 1117 Budapest, Dombóvári str. 26., Hungary), RAGT (Budaörs, story 3, Keleti str. 7, 2040, Hungary), Syngenta (Budaörs, story 3, Keleti str. 7, 2040, Hungary), etc. In the project, 23 hybrids were tested. Therefore, we agreed that, in the first two years, 18 will be identical, with a possibility to include five new hybrids in the second year. The same was true for the second two years. However, four controls were tested across the four years. Trials conducted for this study were part of the Trials System of Hungarian Maize Club Association (a public organization), which aims to improve the knowledge of farmers. 

The experiments were conducted in Kiszombor, 25 km east of Szeged, in the Maros River Valley (GPS coordinates: 46°12′49.0″ N and 20°09′57.9″ E). This is a research station of Cereal Research Ltd. Yearly precipitation varies between 350 and 1100 mm, and soil pH is 6.98. The latest soil humus content was 2.21%, which has been decreasing for a long time; the NO_3_-N is 5.8 mg/kg, classified as extremely low; the P_2_O_5_ is 280 mg/kg, the K_2_O is 317 mg/kg, the Mg is 376 mg/kg (all three are at high rate), Zn and SO_4_ are poor, and the rest are moderate. Therefore, in autumn, 160 kg of Genesis (8:21:21% NPP rate) was administered, and 80 kg of Nitrosol (46% carbamide) was administered in the spring. Both are products of Péti Nitrokomplex Ltd. (8105 Pét, Hősök square 14, Hungary). Irrigation was performed (40 mm per treatment) when necessary, just after sowing (between 25 April and 3 May) in mid-June to enhance ear differentiation and at the end of the inoculation time in the third week of July (or somewhat later depending on the weather). To control the European corn borer, 0.2 L/ha Decis (Bayer Inc. Leverkusen, Germany a.i. deltamethrin 50 g/L) was used (1 or 2 treatments). For weed control, Lumax from Syngenta (5732 Mezőtúr, Hungary a.i. 37.5 g/L mesotrione, 375.0 g/L S-metolachlor, and 125.0 g/L terbutaline) was applied at rate 4.5 L/ha, Dezormon from Nufarm Hungaria Ltd. (1118 Budapest Hungary, a.i. 600 g/L, 2,4-D), and Shadow 200 from BASF (BASF Hungaria Ltd., 1132 Budapest, Vaci str. 92, Hungary, a.i. 200 g/L dimethenamid-P, 200 g/L metazachlor, and 100 g/L quinmerac) at a dose of 2.5 L/ha were used, depending on the availability and weed composition. 

Each plot consisted of four rows 8 m long. At half distance, a 50-cm-wide road was cut out, and the length was dissected into two 3.75-m-long parts. In the lower part of the row, Isolate 1 was used, and the upper part was inoculated by Isolate 2. The isolates for the three rows were: *F. graminearum* (No. 3 and 4), *F. verticillioides* (No. 1 and 2), and *Aspergillus flavus* (No. 1 and 2). The fourth row was the control without artificial inoculation to allow us to observe the background ear rot severity and toxin contamination (Table 1). Three replicates (all with four rows) were organized in a randomized block design. Due to the use of two independent isolates, for each hybrid, we had four independent biological replicates in the two years; for the control hybrids, we had eight. This amount of data is sufficient for drawing reliable conclusions. 

### 2.2. Isolates and Inoculation

Two isolates were used for all three fungal species. For *F. graminearum* isolates, No. 3 and 4 were used; for *F. verticillioides*, No. 1 and 8 were used; for *A. flavus*, No. 1 and 2 were used. All were isolated from naturally infected grains. A PCR-based method was used to identify *A. flavus* isolates. Here, a region of the calmodulin gene of the fungal DNA was amplified using the primers cmd5 and cmd6, as described by Hong et al. [60]. For *F. graminearum*, the IGS-RFLP method was used [61]. For *F. verticillioides*, the method of Baird et al. [62] was followed by testing for the presence of the FUM1 gene from the fumonisin gene cluster. For DNA extraction, the methodology of the cited papers was followed [60,61,62]. The strains were deposited in the Microbe Gene Bank of Cereal Research Nonprofit Ltd., which is part of the Hungarian National Centre for Plant Diversity and is freely accessible. Their deposit numbers are as follows: Fg3: NGBAB142629; Fg4: NGBAB142696; Fv1: NGBAB142625; Fv2: NGBAB142624; Af1: NGBAB142601; Af2: NGBAB142602. The nucleotide sequences for species determination were as follows: *F. graminearum* and *F. verticillioides*: EF1-α primers: ef1: ATGGGTAAGGARGACAAGAC; ef2: GGARGTACCAGTSATCATGTT; *A. flavus*: CaM primers: CMD5: CCGAGTACAAGGARGCCTTC; CMD6: CCGATRGAGGTCATRACGTGG. For inoculation, the toothpick method developed by Young [63] and modified by Mesterházy [35] was used measuring kernel resistance. Wood toothpicks were washed three timed to be free of tannins and other fungal growth inhibitors. Once air dried, toothpicks were placed into glass flasks and supplied with a liquid Czapek-Dox medium for an hour. After removal of the medium, except for 5 mm at the bottom of the flask, the mouth of the glass was closed by a cotton cork. They were autoclaved at 120 °C for an hour. After cooling in a sterile inoculation box, 2–3 inocula were transmitted to the wet toothpicks. The toothpicks were ready to use in three weeks stored at room temperature without direct sunshine. As no suspension was used, no conidium concentration could be measured. The aggressiveness of the isolates was tested in previous years on maize ears [5,25]. Silk channel inoculation, which mostly works under cooler and more humid conditions [49], has failed in Hungary [25]. Inoculation was performed for all three pathogens and their isolates 6 days after 50% silking by inserting infested toothpicks into the middle of the upper ear in a hole made by an awl 15-mm-long and 1.5-mm-wide. As most hybrids in Hungary belong to the FAO 200–300 and early 400, and the dry and mostly hot temperature speeds up plant development, we had to make the KR inoculation 6 days after midsilking. Normally, 15–18 ears in a row were inoculated. There was only one inoculation time per row, and late flowering ears were discarded. The toothpicks were left in the ears until harvest, so that the inoculated ears could be identified. 

### 2.3. Evaluation of Symptoms and Risks

For all three pathogens, the same percent scale was used [5,25]. Its origin is the scale suggested by Reid et al. [32], but it should have been modified to differentiate the ear rot severities more clearly for toxin comparison. In a regular ear, 700–800 grains grow. At 1% infection, 7–8 grains show visible infection. When only one grain is infected, the rate is about 0.15%. Rates from above 5% to 100% were considered on a scale by 5% steps. For evaluation, only ears where the mark of the toothpick could be identified were considered. This was to secure improved sampling. This was important because, at higher resistance or lower aggressiveness (such as *A. flavus* and *F. verticillioides*), some ears were not infected, but the trace of the toothpick signaled that the inoculation was performed. All ears were assigned two numbers: the first was the percentage of the visually detectable grains whose infection seemingly originated from the infested toothpick; the other was assigned to grains seemingly infected independently of the toothpick. This latter was considered an additional natural infection. The insect-wounded and Fusarium-infected ears were not considered to avoid mixing artificial infection with the insect damage caused Fusarium and toxin contamination. For correct sampling, this was an important precondition. No such description was found in the literature in our research. 

For the control role, the two hybrids chosen were not perfectly suitable, as these reacted differently to diseases and toxins. Therefore, we related the performance of the hybrids to the arithmetical mean of the 18 hybrids. In the tables the genotypes (low risk) that were lower than 50% of the experimental mean are highlighted in dark green. The hybrids (from a low to medium risk) between 51% and the experimental mean (100%) are marked in light green. By adding 50% of the experimental mean to this mean, the medium- to high-risk hybrids are grouped and highlighted in yellow. All other hybrids were classified as high risk, highlighted in orange. In the naturally infected materials, the toxin limit agrees with the limits of EC regulations [56,57,58,59].

### 2.4. Preparing Samples for Toxin Analyses 

For toxin analyses, we followed our own procedure [25]. Briefly, five maize ears without insect damage and average ear rot values were selected. After an evaluation of ear symptoms (within 24 h), they were placed into a plastic string web Rashel bag and kept in a dry room until dry. Following shelling, the whole amount about of 1 kg was roughly ground to 1–2-mm particles, which were thoroughly mixed. This was conducted separately for all replicates. Afterwards, from each replicate, 100 g were separated, pooled, and mixed again to decrease the sampling error to the lowest possible. From this roughly mixed material, 100 g were separated for toxin analysis and sent to Nagyigmand, an accredited laboratory. As such, the infected particles could be better homogenized for toxin distribution than any other methods used for whole grains. This was useful for *F. graminearum*, but even more so for *F. verticillioides* and *A. flavus*, for which the infection severity of 1% or lower was rather characteristic and therefore much more critical. 

### 2.5. Toxin Analysis

For the analysis, the methodology of the accredited Bonafarm Feed Laboratory Nagyigmand, Hu., was used [64]. All the reagents and solvents were bought from Thomasker. The solvents and glacial acetic acid (AcOH) were bought from Honeywell™. Ultrapure water was made freshly every day using a Human Corp. Zeneer Power I (Human Building 36, Garak-ro, Songpa-gu, Seoul, Republic of Korea, Postal Code 05694) water purification system. Mycotoxin reference standards were supplied by Romer Labs^®^ (Romer Labs Inc., 130 Sandy Drive Newark, DE 19713, USA).

For the preparation of standards, briefly, for AFB1, the RomerBiopureTM Mix 5 was diluted five times to obtain a stock solution. For the toxin analysis of fumonisins, RomerBiopureTM Mix 3 was applied. To produce the stock solutions, a 60-fold dilution was applied. The RomerBioPurTM DON was diluted 66-fold to prepare the stock solution. 

For sample preparation, the maize samples were ground using a Perten Laboratory mill (Type: 3310, Perten Instruments, 126 53 Hagersten, Sweden). Five grams from each sample were separated and placed into 50-mL PP centrifuge tubes. This consisted of 0.5–1 g of the subsamples that were collected from different parts of the 100 g sample. Afterwards, 40 mL of an acetonitrile/water (AcOH, 20:79:1, *v*/*v*/*v*) solution was added to AFB1 B1 and fumonisin B1+B2. For DON, ultrapure water was used for extraction. The mixture was vortexed and extracted by a rotary shaker for 1.5 h at 200 shakes/minute. Following centrifugation at 5000 rpm for 15 min, the supernatant was passed through a PTFE filter (0.2 mm pore size). 

For chromatography, a Kinetex^®^ C18 100 Å UPLC column (2.1 × 100 mm, 1.7 μm) was used and kept at 30 °C. The flow rate was 0.35-mL min-1, and a 5-µL partial loop injection was used (a loop of 20 µL). Mobile phases were buffered with 5 mM of ammonium-acetate. Ultrapure water with 1% AcOH and 4% MeOH was used as Mobile Phase A, and MeOH containing 1% AcOH with 2% ultrapure water was used as Mobile Phase B. 

UPLC-MS/MS was performed in MRM mode on an AB Sciex QTRAP^®^ 6500 tandem mass spectrometer connected to a 1290 Agilent Infinity II-UPLC system (Agilent Technologies, 5301 Stevens Creek Blvd. Santa Clara, CA 95051 USA) equipped with Agilent components as follows: a G1316C Thermostatted Column Compartment, a G1330B Thermostat, a G4220A Binary Pump, and a G4226A Sampler.

The main mass spectrometer worked at a source temperature of 350 °C, a curtain gas of 40 psig, an ion source gas of 35 psig, a detector voltage of 5 kV, and an entrance potential of 10 V. The collision gas was set to medium. MS/MS conditions were optimized in the Analyst^®^1.6.2 compound optimization module using the direct infusions of each analyte’s standard. The detailed parameters of each analyte were optimized, as summarized in Table 2. FB_1_+B_2_ and AFB1 were measured positively. For DON, the negative mode was applied.

For calculations, all the sample solutions were measured against a calibration curve for the analyte of interest. Raw results were calculated using the Analyst^®^ 1.6.2 software package. The peak concentration was calculated as cspike = cstockˑ(Vspike÷Vfinal). The results were corrected with the recovery of the standard addition, i.e., the weighed sample, and the extraction solvent was added using the following formula: csample = (crawˑVextr ÷ msample)/((crawspiked/craw) ÷ cspike).

### 2.6. Statistical Methods

For the artificial ear rot data, the four- and two-way ANOVA models were used in three replicates. For the toxin data, the two-way ANOVA model was used without replicates from the Excel Analytical Tools. As such, the year and isolate effects could be balanced to some extent. For stability, the variance from the one-way ANOVA (Excel) was used. Additionally, regression and correlation tests were used to compare ear rot and toxin data using the built-in Excel program. For the four-way ANOVA, first a two-way ANOVA was made by Excel from the sums of replicates, and the four-way ANOVA was then conducted by the functions presented by Sváb [65] and Weber [66]. Their statistical tables were also used to evaluate significance levels. The calculation of the overproduction of toxins is demonstrated in Figure 1, following the procedure of Mesterhazy et al. [25]. This was followed for all further figures of this type. 

## 3. Results

### 3.1. Experiment 1, 2017/2018

#### 3.1.1. Ear Rot Severity

The two-year data of the 18 hybrids showed rather variable results (Table 3). The means of the two isolates showed somewhat closer correlations than the yearly data separately. *F. graminearum* caused the most severe symptoms compared to the other two species, which demonstrated rather low infection severity. For this reason, the LSD 5% values were calculated separately for each fungal species and the control. The differences were significant against all pathogens. The risk was calculated separately for each toxigenic species and the control. In an ideal case, the resistance (based on visual primary symptoms) would be colored dark green for all pathogens, but such a hybrid was not found. However, four hybrids were identified with lower ear rot severity than the experimental mean (highlighted in dark and light green). Their names are printed in bold. In other cases, one or more traits are marked in yellow or orange, indicating a higher epidemic risk. In one case, all responses are marked in orange, e.g., highly susceptible, indicating risk to all pathogens. The ranks were also calculated, and their means are also shown. The variance for the ranking is also given to show the stability of the responses. We need stable, lowly infected plants. Correlations between hybrid data and different ear rots are not close and, in several cases, not significant. Here, Szegedi 521 should be mentioned, as it was among the best for the three ear rots, but it showed a higher infection at the control rate. 

The natural Fusarium severity levels were low; in each artificially inoculated treatment, they remained under the control level (data not shown). Natural Aspergillus infection in the control was found only in one hybrid. In 2017, six genotypes were identified from the 18 hybrids in 2018 (data are not shown in detail). 

The ranges of the resistance levels (Table 3) showed a similar picture. We only observed two genotypes with dark and light green colors, and two were found at high risk to all traits. Szegedi 521 showed resistance to all fungi with artificial inoculation; therefore, further testing was required. For the rankings, the variance was also considered; this shows the stability of the reactions to the different traits. Korimbos and P9240 had extremely low rankings to all traits. DKC 4541 and PR37F80 had high rankings, both above 15. The remaining 13 genotypes showed rather wide variability to the different toxigenic fungi. The correlations were similar with two differences: In the rankings, *F. graminearum* resistance did not correlate with *F. verticillioides* and *A. flavus* resistance, and the correlation between individual traits and means showed a much more uniform response compared to the data shown in Table 3.

The ANOVA showed highly significant main effects (Appendix A). From the interactions, the two-way interactions were important and involved the maize genotype. The genotype/toxic species interaction was significant, indicating the different responses of the hybrids to the different pathogens. The influence of the year on the performance of the toxigenic species was highly significant. No significance was found for the genotype/isolate and genotype/year interactions, e.g., the variety reactions were rather stable. The three- and four-way interactions are not easy to explain, but their influence (MS and F values) was low, only significant in some cases. As the difference between toxic species was large, the interactions containing this source of variance were unusually high and significant. This was the reason for presenting an ANOVA for all fungi and control separately (Table 2).

#### 3.1.2. Resistance to Toxins

The three replicates for isolates of the given year were pooled. The toxin data (Appendix A) showed the highest toxin contamination for DON (mean = 14.6 mg/kg), FB_1_+B_2_ had a medium position (mean = 3.0 mg/kg), and the lowest concentration was found for aflatoxin with a mean of 119 mg/kg. The total aflatoxin contamination limit is 20 μg/g in the animal feed and 4 μg/kg in the human food; numbers are listed in Table 4 in this form. For each toxin, the LSD 5% values were significant. The correlations between the two isolates in a year correlated significantly with fumonisins in 2017 and 2018 and with aflatoxin in 2017. The same isolates in the two years correlated significantly for DON Fg3 2017 and Fg3 in 2018. For FUM B_1_+B_2_, no clear tendency was found. The correlations between isolates and years were not significant, except one. For AFB1, only the 2017/2018 correlation was significant between Af1 and Af2. The hybrid reactions to the two isolates in the same year were widely variable. Two different isolates, or the same isolate in two different years, can behave differently in the same year. This is also an argument supporting the use of more parallel isolates to have more data for reliable decisions.

Importantly, for each toxin, we identified hybrids that produced stable responses across years and isolates at a low toxin level, e.g., lower than the means; their data are printed in bold. Three hybrids were positive variants across all toxins: DKC 4717, DKC 4943, and Cardixxio Duo. For aflatoxin and fumonisin, DKC4541, DKC5542 were positive examples, but their DON response was higher than the mean. Only the hybrid P9911 was found with high susceptibility to all toxins. 

The comparison of ear rot and toxin data (Figure 1) showed only a loose correlation for *F. graminearum* and DON (r = 0.4289, *p* = 0.1). For *F. graminearum*, one DON overproducer (Valkür) was identified. Valkür had a 23.54-mg position in the linear function. The distance between the measured value (63 mg/kg) and its place on the regression line (23.54 mg/kg) was 39.7 mg/kg, and this is larger than the 20.16 mg/kg LSD 5% value. The overproduction for Valkür was proven. Korimbos was close to becoming a DON overproducer. Without Valkür and Korimbos, the correlation increased to r = 0.71 (*p* = 0.01), indicating that most genotypes responded proportionally to the two traits. In conclusion, for most hybrids, an agreement between visual symptoms and DON was found.

For *F. verticillioides*, a significant positive correlation was found between ear rot percentage and FB_1+2_ (r = 0.5908, *p* = 0.01, Figure 2). Two hybrids (P9911 and 4517) showed toxin overproduction and had a significantly higher fumonisin content than was forecasted based on the infection severity. Without them, the correlation increased to r = 0.69 (*p* = 0.01), indicating that most of the hybrids had similar ear rot and toxin responses. Six hybrids had low FER infection and fumonisin contamination. Ten hybrids belonged to the low ear rot and low fumonisin content group. The conclusion is similar: the majority of the hybrids had proportional disease severity and toxin responses. 

*A. flavus* (Figure 3) showed exceptionally low ear rot severity, but the AFB1 contamination differed. Five genotypes produced significantly more AFB1 than forecasted by the regression line. For these genotypes, the difference between the data point and the corresponding point on the regression line was larger than the LSD 5% value for AFB1, i.e., 66.29 μg/kg (right from Silo Star in Figure 4, between 77 and 214). Therefore, their aflatoxin overproduction was confirmed. They cause an additional food safety risk. Six genotypes had a low infection and a low AFB1 concentration. Excluding the five correlation breakers from the correlation, an r = 0.60 value was computed as being significant at *p* = 0.05. This means that most of the genotypes in the FER and FUM data correlated similarly. 

The table summarizing the resistance and toxin resistance of the hybrids ranks them according to their resistance to *F. graminearum* ear rot (Table 4). The FAO numbers are also included. Three hybrids were identified with low or low-to-medium risk for all traits (names are printed in bold). DKC 4590 showed higher fumonisin contamination, but the very low aflatoxin contamination is valuable. We received comparable data for P9537 and P9241 as well. They belong mostly to the suggested group. Cardixxio Duo had a somewhat higher AER than the mean; however, for AFB1, it was among the bests. This is an example where a toxin is more important than a symptom, and both artificial and natural AFB1 data scarcely support each other. The other hybrids had highly diverging data, often with high-risk designations, but no hybrid could be found with a total high-risk classification for all traits. Low and remarkably high toxin values also co-occurred, such as in Valkür. The DKC 5542 produced exceedingly high AFB1 with artificial inoculation and high natural contamination. It had low values for DON and fumonisins via artificial inoculation, which may be different among the other toxins. When natural and artificial data agree, they support each other; the opposite indicates a more problematic case. For Korimbos, the resistant control had low infection numbers against all toxigenic species but showed 298 μg/kg AFB1 in artificial infections and 33 μg/kg in natural infections. The latter is proof of a preharvest origin. This is valid for all hybrids with any AFB1 concentration. This is the first proof of the presence of AFB1 in Hungary due to natural preharvest infections. It is important that 11 hybrids including the control Korimbos were free of visible Aspergillus infection but had an aflatoxin contamination level higher than zero. From the coloration of the cells, we can infer detailed information about the resistance behavior of the hybrids.

The correlation matrix allowed several important conclusions to be drawn (Appendix A). *F. graminearum* correlated positively with *A. flavus* ear rot severity, but negatively with AFB1 contamination, and no correlation with FAO numbers was found. This indicates that the regulation of ear rot and AFB1 contamination might be different. *F. verticillioides* behaved more consistently: Ear rot correlated positively with fumonisin content and significantly with natural fumonisin contamination. Comparably, *A. flavus* ear rot correlated positively with *F. graminearum* and *F. verticillioides* ear rot response, but not with natural Aspergillus infection. DON contamination correlated significantly with FAO numbers, e.g., later hybrids tended to contain more toxins in this test. Natural Fusarium ear rot correlated positively with artificial and natural fumonisin data; in this case, the correlation system seemed to be the most consistent. Fusarium ear rot tended to be higher in later hybrids, but the correlation was just above the limit. In Aspergillus, however, the FAO numbers had a significant negative correlation with ear rot data, e.g., later hybrids were healthier, but contained more AFB1. This is a contradiction we have previously observed. From the possibly 66 correlations, only 11 showed significances. The matrix clearly shows that the Fusarium and Aspergillus symptoms and toxin contamination are highly sensitive and complex phenomena. Based on correlations, forecasting the toxin or infection rates, and finding cross resistance to different pathogens are not possible. Several significant correlations indicate that *F. verticillioides* and *A. flavus* agreements might be present but are not generally valid. The earliness or lateness yielded the surprising result that a hybrid tends to have less disease later (natural Fusarium coverage and DON concentration). In other traits, no significant influence was detected.

The reductions in the disease or toxin content compared to the most susceptible genotype (Appendix A) were highly significant. The mean reduction in visual symptoms (the mean of hybrids) was 45%, 67%, and 60% for Fg, Fv, and Af, respectively, and the mean reduction in toxin contamination was 73%, 78%, and 68%, respectively. The maximum reduction in ear severity was 79% for *F. graminearum,* 97% for *F. verticillioides,* and 94% for *A. flavus*. The maximum reductions in toxins were 88% for DON, 93% for fumonisins, and 95% for aflatoxin B1. However, the hybrids varied widely. The minimum reductions in symptoms were much lower: 22% for symptoms caused by *F. graminearum*, 5.3% for symptoms caused by *F. verticillioides*, and 17% for symptoms caused by *A. flavus*. For natural infection, the mean reduction in Fusarium infection was lower, but it was high for *A. flavus* (at low infection severity). The reductions in the maximum toxin percentage for natural *F. graminearum* was 100%, 99% for fumonisins, and 98% for aflatoxin B1. It seems that the toxin reduction was larger than the decrease in visual symptoms, providing a further argument for assigning a larger weight to the toxin contamination in the risk analysis. This finding has scientific importance as well.

### 3.2. Experiment 2, 2019–2020

#### 3.2.1. Ear Rot Data

In this test, again, 18 hybrids were screened. Among the two tests, four of the hybrids were common. The testing methodology was the same as in the first experimental series. As *F. graminearum* was the most aggressive, the LSD 5% values were counted separately for all toxigenic fungi (Table 5). As *F. graminearum* dominated the means, the LSD 5% values were counted for all toxigenic species separately as well as for the check. The color use followed the scheme applied in Table 3. For resistance, six hybrids were found with a low or low-to-medium designation (dark and light green); in the controls, the value were rather low, and yellow performance was acceptable when the difference between limit value and the given hybrid was less than the LSD value. H15 is just on the border. We found no significant correlations between toxigenic species, and only the *F. verticillioides/A. flavus* relationship was significant. Natural infection showed a significant correlation with *F. verticillioides* and *A. flavus*. This differs from the first test and provides another example that the different hybrid sets may behave differently. As the mean data were dominated by *F. graminearum*, for the stability tests, the variances of the one-way ANOVA from the artificially inoculated treatments and the naturally infected control data were used. The genotypes having low means and low variance were Korimbos, P0725, Koregraf, ES Lagoon, and Armagnac. The highest stability was shown by P9415, which was susceptible to all traits tested. The correlation between *F. verticillioides* and *A. flavus* was significant, and both had a significant correlation with the natural control data, indicating that this situation is not accidental. 

The ANOVA indicated highly significant main effects, except for the two isolates that were closer to each other than in 2017–2018 (Appendix A). The toxigenic species were similar and highly significant, and the two years differed significantly in epidemic strength. The genotype/toxigenic species interaction (A × B) was significant. *F. graminearum* did not show a correlation with resistance to the other two species. However, a significant relationship was detected between *F. verticillioides* and *A. flavus*. Therefore, it may not be coincidental that the infection severity in the control showed significant relationships with the response to the two species. As the correlation between individual toxigenic species and their mean was dominated by the *F. graminearum* data, the other correlations were not significant. This was not a surprise. The year/genotype interaction was significant, but not at a high level. The toxigenic spp./year interaction was highly significant, which was expected from the yearly differences. 

#### 3.2.2. Resistance to Toxin Contamination

The toxin data are presented in Appendix A. The DON values were the highest (mean = 48.4 mg/kg), and their extent was significantly higher than in the first test. The fumonisin data were about the same, and the lowest values were found for aflatoxins. The experience with isolates and years were similar, but the contradictions were more expressed. Amongst the four data series of the toxins, no significant correlations were found. For each toxin, we found data pairs. For Fg4, Sy Zephyr had 193 mg/kg DON; in the following year, that concentration was only 5.4 mg/kg. The ES Lagoon Fg4 isolate produced 29.6 mg/kg DON in 2019, but this increased to 105.3 mg/kg in 2020. For aflatoxin, Af1 produced 143 μg/kg in 2019 in ES Lagoon, but the same isolate produced 3261 μg/kg in the following year. For Sy Zoan, the two *A. flavus* isolates were found at 2 and 15 μg/kg in 2019 and at 69 and 4947 μg/kg in 2020, respectively. The variances can help identify the genotypes with low means and low variance. As such, for DON, seven genotypes were found with a low amount and low variance. For fumonisins, nine genotypes were identified, and five were found for AFB1. It seems that these resistances to the different toxins do not correlate with each other; however, in four hybrids (whose printed names are in bold), the three resistances to toxins seemed to agree. Therefore, the necessity of parallel testing the hybrid to different toxigenic fungi is now well supported. Given the 18 hybrids, this is an acceptable rate. Two hybrids were identified, and the use of the two isolates significantly helped to achieve more reliable results. The data clearly show that the resistance to toxigenic species and their toxins is highly sensitive and variable; therefore, a larger database is needed to find appropriate answers and make suitable decisions. The correlations between the traits tested were mostly not significant (Appendix A). It is clear that this correlation system cannot provide a basis for resistance screening or breeding.

For kernel ear rot and DON regression (Figure 4), two genotypes with doubled DON production were identified, and another two were found with significantly less DON production than anticipated from the regression line.

The DON mean differences were high; however, due to the variability in the basic data, the genotype differences were not significant (Appendix A). Even the correlation between infection severity and toxin was significant (r = 0.5078, *p* = 0.05). The variation was rather high; for example, at 22% ear rot severity, the DON values varied between 7.3 and 101 mg/kg. Therefore, feed safety risk estimation without toxin control is not possible. However, in this case, six hybrids with low GER and DON were also identified. By removing the correlation-breaking Korimbos and P0718E from the line, the correlation improved to r = 0.68 (*p* = 0.01), indicating the significance of several off-type hybrids in forming the correlation. 

For fumonisins (Figure 5), the situation was similar: two hybrids strongly broke the correlation between low infection severity and high FB_1_+B_2_ concentration. The correlation was not significant. Without these two genotypes, the correlation improved to r = 0.472 (*p* = 0.1). The situation was similar to that observed for the GER/DON relationship. Additionally, here, eight hybrids were also identified as having low infection severity and low toxin contamination. 

The AFB1/*A. flavus* regression showed wider genotype differences: Six hybrids belonged to the correlation-breaking group (Figure 6). As such, no significant correlation was possible. Without the five correlation-breaking hybrids, the correlation increased to r = 0.5788 (*p* = 0.05), indicating that the majority of the genotypes reacted similarly to disease and toxin production. As it is not known before testing which hybrids would react proportionally and which would not, all hybrids were measured. Of course, there was a possibility that a hybrid with higher Aspergillus infection than average was discarded. From the 18 genotypes, 8 were identified with low infection severity and low toxin contamination. However, the visual infection severities were exceptionally low, and even the most infected had only 0.6% severity; at this level, nobody would discard any hybrid. This logic is sounder for GER, but every genotype must be measured for AFB1. 

Figure 7 summarizes the symptoms. The highest infection severity was found with *F. graminearum*, from nearly total susceptibility to a lower than 10% mean severity. The optical sight is often higher than the reality is, as the backside of the ear is normally much less infected or not at all. However, at evaluation, the whole ear surface is considered. The Aspergillus infection is quite severe in the first two hybrids, bur for C and D hybrids, this is much lower, most ears are symptomless, or one or two infected grains can be identified. For *F. verticillioides*, Hybrid B is the most susceptible, but Hybrid D is close to symptomless. 

It seems that, in highly epidemic years, the data have a higher variability. However, in each case, we could identify hybrids with low infection severity and low variance. 

Table 6 summarizes the 2019–2020 mean data across isolates and years. The FAO numbers are also included to show the possible influence of earliness or lateness. The risk classes can be easily identified by the different colors. Not one hybrid was found with all data in the dark green, most resistant, group. ES Harmonium and Talisman have the lowest values considering all columns and are somewhat higher for natural infection, but the toxin data are good. For Konfites and Kleopatras, all colors can be found. The same variability was found for toxin contamination from artificial inoculation. An excellent low Aspergillus infection was identified in P0725 at an extremely high AFB1 contamination at both inoculations. The hybrid Sy Zoan was found to have the highest risk to all toxins, but in terms of resistance, two are marked in light green. We observed similar phenomena in the control values. With no visual Aspergillus infection, a AFB1 concentration of many hundred micrograms per kilogram was measured (Korimbos). Conversely, extremely high artificial DON contamination was found for P9718E, but all values were excellent with natural infection. This is a contradiction. In Koregraf, however, all artificial and toxin data were good, but the natural AFB1 concentration was extremely high. These data are the basis of the risk evaluation in Section 4. 

Of the 66 correlations from 2019–2020 in Table 6, only 8 were significant (Appendix A). In terms of symptoms, *A. flavus* ear rot correlated significantly with *F. verticillioides* resistance data. However, in the toxin data, this was not true. Artificial DON data correlated positively with FAO numbers, as found in the previous experiment. The same controversy was found for the Fusarium check and FAO numbers. The general tendencies show so far that only a minority of the correlations are significant, but most cases are not the same. Both tests support the view that a stable correlation matrix that would support the use of automatic models to select hybrids more resistant to ear rot or toxin contamination does not exist. We think the main source is the different genetic background of the hybrids. Therefore, we should be careful when drawing general conclusions on maize from tests with even 20–40 hybrids. 

These findings suggest that we must select hybrids that do not have a high-risk classification. DKC 56,830 had a high value at natural DON contamination but a value close to zero with AFB1. In dairy, DON and FB_1_+B_2_ could be more flexible than AFB1 (limit: 20 μg/kg). The hybrids marked in bold are suggested as lower-risk hybrids for commercial production. Additionally, we must consider Talisman’s and Korimbos’ 11.5% and 13.5% infection severity, respectively, of *F. graminearum* and the 7.5 and 76 mg/kg DON, respectively, in their grains. Making an appropriate decision without toxin data is not possible. It is important that all natural toxin data are obtained from preharvest, so that their preharvest origin is proven. 

The reductions in disease and toxin content due to higher resistance were different (Appendix A). The mean reduction in ear rot severity (kernel resistance to disease) was 38.4%, 74.5%, and 60% for *F. graminearum, F. verticillioides*, and *A. flavus*, respectively, and with maximum reductions of 71%, 94%, and 91%, respectively. The means for the toxin reductions (DON, FUM, AFB1) were 56%, 60%, and 76%, with maximum values of 93%, 87%, and 97.1%, respectively. The numbers for ear rot and toxin reduction following artificial inoculation were closer to each other than they were in 2017–2018, but the reductions in DON and AFB values were also much higher than found for the symptoms. 

The data for natural ear rot caused by *Fusarium* spp. and *Aspergillus* spp. had a 45% and 76% mean reduction, the maximum was for *Fusarium* spp. at 82%, and many hybrids showed 100% reduction with aflatoxin. DON, FUM, and AFB1 showed reduction means of 77%, 75%, and 91% with maximum values of 100%, 98%, and 100%. We observed, for example, for GER in Valkür an 18.6% reduction in ear rot symptoms and an 18.2% reduction for DON. In Sy Zephir, the two values were 4.1% and 71.9%, respectively, and many other examples could be cited. This means that resistance should play an important but not an exclusive role in the reduction of toxin contamination, but different genetic mechanisms can be supposed to be at play. Similar findings to artificial infection were obtained for natural infection and toxin contamination. It seems the toxin numbers cannot be predicted from the natural infection data, so every hybrid should be tested. 

Appendix A shows that the same ear rot values may cause different toxin contaminations in the two tests. Therefore, the rates of visual ear rot data and the toxin contamination were counted for a 1% visual ear rot infection (Toxin/%). The mean rate between the maximum and minimum toxin contamination for a percentage of visual infection was in 2017–2018 4.84%, 8.83%, and 9.64-fold% for *F. graminearum*, 17.61% for *F. verticillioides*, and 56.14% for *A. flavus*. The same numbers were in 2019–2020 9.64%, 17.61%, and 60.59%-fold, respectively. The highly aggressive *F. graminearum* showed lower rates, with the differences being two-fold higher for the less pathogenic *F. verticillioides* and six-fold higher in the least aggressive *A. flavus*. This means that the visual scores alone are not useful for predicting food safety risks. For highly aggressive toxigenic species, the risk is lower, but for the less aggressive species, the risk of the probability to select false positives or false negative results is much higher. In five hybrids, we found low rates and high stability (low variance) for all three toxigenic species, 5 of the 18. Korimbos had the highest rate with respect to all toxigenic species; even in resistance classification, it was among the most resistant genotypes. This shows clearly what consequences may come when only resistance to ear rot data is considered in the hope that toxin rates will also be low. The genetic basis is not known. However, it would not be a large surprise if genetic effects are found. 

Over the four years, four hybrids were tested each year (Table 7). Their mean data across four years and two isolates (eight independent data sets) showed a three-fold DON/GER rate difference between genotypes. For FUM/FER, the difference was two-fold; for AFB1/AER, the rate showed an 11-fold difference. Korimbos had the highest rates, as shown in Table 7. The susceptible control DKIC 4511 had significantly higher ear rot values, but except for FUM, the toxin/ear rot rate was significantly lower. It seems that this data set can be collected in two years in two locations. This is thought to be the minimum data set that would allow for a fairly solid risk estimation. 

## 4. Discussion 

### 4.1. Evaluating Hybrids under Artificial and Natural Infection Regimes

When we started artificial inoculations in maize nearly 50 years ago, the breeders requested that we reproduce the symptoms they experienced with natural infection. Over the decades, it became clear that this request could not be fulfilled for many reasons [8]. Though we concluded that natural infection and toxin contamination were not useful for selection and breeding purposes, they are not useless data. At harvest, the preharvest contamination should be measured. When it is higher than the official limits, the grain can only be marketed at a lower price, if at all. These data help to store separately the differently contaminated grains. The data can serve for feedback to breeding under an artificial inoculated selection. Databases on natural toxin contamination are needed to decide on breeding or selection programs, as this is the most effective method of control [2,8,67]. The three toxins occur together in Africa [68], but no resistance tests have been conducted on them. This situation may also occur in other maize-producing regions. This should be changed. 

Most authors use an inoculum (pure isolates and mixtures) [32,69,70,71]. The two isolates used had a positive impact; our data support the finding that the isolates differ not only in their infection severity but also in the toxin contamination caused [55,72]. The same isolate may behave differently in different years. Earlier, we obtained similar data [50], but the toxin responses for DON could not yet be tested at that time. However, the GER data clearly show differences in response to the different isolates. The conclusion was that more isolates provide a more reliable picture. As registration normally requires two years, by using two or more isolates, significantly more data can be produced to assist reliable decision-making. Looking at the four control hybrids across the four years (Table 6), we think that this amount of data secures a solid basis for decision makers. It is similarly important that the toxin samples should be obtained from carefully planned and performed tests; the representativity as well the sampling of the experimental material for toxin evaluation should be secured. Otherwise, the resistance and toxin data cannot be compared and evaluated correctly, and a correct risk analysis cannot be performed. 

The close relationship between the resistance to disease and the resistance to toxins for *F. graminearum* and *F. culmorum* has been shown [5,25,31]. As a similar result was reported recently [25]; we concluded that there is no reason to test their resistance separately. The only reason for using *F. culmorum* is its lower temperature requirement; therefore, in cooler summers, higher infection rates can be achieved than working with only *F. graminearum* [73]. 

No study in the literature deals with the resistance to, and the toxic behavior of, *F. graminearum, F. verticillioides*, and *A. flavus* together, except our recent study [5]. In this paper, the aflatoxin could not be measured because the toxigenicity tested on rice did not work on maize. It seems that the test on corn ears is an improved solution. However, a RT-PCR method may help to identify false positive isolates [23]. In this test, the correlations between symptoms (resistance) with respect to *F. graminearum, F. culmorum*, and *F. verticillioides* (not *A. flavus*) were mostly significant, but this was not true for toxin production. 

The presented results agree with earlier literature showing that complete agreement with the three different toxigenic fungi does not exist. Agreement between two of them support this result. For example, the agreement between *F. graminearum* and *F. verticillioides* was only partial; significant correlations in resistance have been found [35,38,39,41], but no significant toxin correlations were provided. For *F. graminearum, F. culmorum*, and *F. verticillioides*, the results were similar [25] with good correlations between visual scores (resistance), but this was not valid for the toxins. Löffler et al. [40] did not find agreement between ear rot and toxin levels (*F. graminearum* and *F. verticillioides*), so they suggested separate resistance control against the two pathogens. Now we can add *A. flavus* to this list. The agreement depended on the hybrid population tested, but it was mostly not significant in toxin relationships, as shown by our data. Our earlier tests [5] did not show any correlation between *F. graminearum* and *A. flavus*, but a correlation between resistance to *F. verticillioides* and *A. flavus* was occasionally presented. The finding of Rose et al. [13] supports this view. In this study, we found several positive correlations between kernel resistance in the two species, but this was not valid in the toxin data. About 10–20% of the tested hybrids belonged to the low to medium risk group against all toxigenic species, 5–10% had a very high risk for all pathogens, and the rest showed variable risk enrollment. For the hybrids having consequently low symptom severity and toxin contamination, we cannot say anything about the genetic background, but based on earlier observations, a pleiotropic effect in several genotypes cannot be excluded. We can suppose that different QTLs cause resistance to different fungi present in the plants at the same time. At present, we can measure the amount of resistance and the amount of toxin production, so the genotypes with low risk can be identified and suitable hybrids can be selected. We are sure that further genetic work will be necessary to identify their genetic background. 

The relationships between resistance and toxin contamination are another key issue. The literature mentions stronger or weaker correlations between ear rot severity and toxin contamination for the three toxigenic species. As the data show from the present experiments, in most cases, significant correlations were found, but almost every set of hybrids and toxigenic species showed entries with very high or low toxin contamination compared to ear rot severities. We referred to this as toxin overproduction or relative toxin resistance [5,25], which was also proven in these tests. Its extent is less for *F. graminearum*, higher for *F. verticillioides*, and highest for *A. flavus*. Excluding these toxin overproducers, the closeness of the correlations significantly increased, indicating that the vast majority of the genotypes have proportional kernel severity and toxin contamination. 

The artificial inoculation results for the kernel resistance and toxin contamination data can be summarized as follows. There are highly significant kernel resistance differences between hybrids that were not selected expressively for resistance or low toxin contamination to any toxigenic fungi. As most breeding firms do not provide such information, this fact supports our hypothesis that effective selection work is not the case. The general lack of a comprehensive correlation matrix does not enable one to select against one fungus such that the resistance to disease and toxins would be automatic to the others. Between visual kernel data, we see significant correlations, but this is not valid for the toxins [5,25,35]. Therefore, the resistance and toxin response should be tested in all cases, and the resistance to the different toxigenic fungi should also be tested separately, as was done in this study. 

Notably, a significant reduction due to resistance occurred in many hybrids but at different rates. The mean reduction was lower (about 10%) for the ear rot rate than for the toxins. This is an important argument for estimating of toxin contamination at a higher significance than that of the visual symptoms in risk analysis. The data support the conclusion of Focker et al. [52] that a significant reduction in symptoms occur due to resistance. The maximum toxin decrease was about 80% for DON and reached or surpassed 90% for the best hybrids. Due to the lack of a solid correlation matrix between traits, the experimental data should be calculated for each hybrid, pathogen, and toxin separately, and we can then provide a detailed risk analysis. 

At present, we detected genotypes with good resistance to different toxigenic species and their toxins. The genetic differentiation of susceptibility and partial resistance is possible, and the existing QTL analyses show examples for this. As such work was not done in this study, there is a necessity for such studies in the future. Additionally, the combining ability of the inbreds, both for yield and resistance to disease and toxins, should be tested. The QTL analyses cannot be used for this purpose. 

The first conclusion is that we have large and highly significant differences in the resistance to ear rot infection or in the toxin contamination of the three main ear rot pathogens, but there are many exceptions. This variability can be exploited to identify and select hybrids with multiple resistances to toxic agents and their toxins. The second conclusion is that there is no generally valid complex resistance to different fungi. Hybrids were identified with good resistance to all three fungal species and their toxins, but their genetic background is unknown. We should say that this is easier than breeding a higher-yielding genotype that is the most complex existing polygenic trait. This is supported by the continuously increasing number of better yielding hybrids. 

### 4.2. Visual Ear Rot, Rachilla-Mediated Infection, and Toxin Contamination

The controversial low visual ear rot values and high toxin contamination require an explanation. The spread of the fungus following inoculation in the ear was more rapid on the rachilla (cob surface) than on the ear surface with seemingly healthy grains [30]. We found a 12–28% difference between the moisture content of the grains and rachis for different hybrids, where the grain moisture varied between 24% and 56% [31]. Christensen and Kaufmann [74] reported that, at 23% grain moisture, the visual spread of the fungus on the cob surface stopped. This is the first report of this problem with artificial inoculation. This phenomenon is shown in Figure 8 for a natural infection, where a significant part of the kernel tops is healthy, but the rachilla-originated germ infection is severe. The visual ear scoring on the unshelled ears does not recognize this infection. For this reason, such hybrids may cause additional toxic problems in silo maize or corn–cob mixtures [30]. Other authors [26,27] have published similar results for *A. flavus*. Wit et al. [75] reported this finding for *F. verticillioides*, and a gene was identified for cob resistance, inhibiting infection from the rachilla [29]. The progression of toxin transformation to AFM1 is possible within the plant, but the information is sparse [75]. 

### 4.3. Advantages of the Suggested Testing Methodology

It is possible to screen registered hybrids, variety candidates, or inbred lines for complex resistance to diseases caused by toxigenic fungi and their toxin contamination without knowing the genetic background. For us the most important task was to receive reliable data on ear rot and toxin resistance. As the test takes two years and the most susceptible hybrids will not go into commercial production or can be withdrawn, a third-year test may be needed. As Table 6 shows, in two locations with three isolates, this task can be performed well by favoring the lower- and low-risk hybrids. As has been stated, the resistance to different toxigenic species should be tested separately. In the control plots, every year, a multitoxin presence was identified at a changing proportion. A multitoxin method is necessary for research, breeding, and the food and feed industry. Initially, these three fungal species and their most important toxins are sufficient. 

The different resistance reactions of the hybrids to different isolates gives the background on the use of more isolates independently at the same time. It is supposed that the different isolates will yield very similar results. However, this is not so. Nobody has shown hybrid or inbred specific races in these fungi. This has also been concluded in the wheat research [76]. As shown in Figure 9 [50], any of them can be chosen as the isolate being assessed. On the other side, H2 and H7 hybrids show very diverging responses to different isolates. According to the literature, these questions have no reliable answers, but the mean of the isolates used could be a solution (the thick red line for the mean). This is why, in previous studies as well as the present study, two isolates were used, increasing the reliability of the data. Of course, two isolates imply a doubly high cost. We made a price estimation: For 200 hybrids, a test with three isolates would cost, with all toxin measurements, about 300,000 USD. In a highly epidemic year, the loss can be 100 million USD. We think this is an economic investment. 

### 4.4. Food or Feed Safety Risk and the Toxin Production for One Percent of the Ear Infections Analyzed

Until now, food and feed safety risks have been evaluated by the visible natural infection, mostly without toxin control. This must be updated.

The following data are necessary to measure food and feed safety risks:Separate artificial inoculation ear rot data for the three pathogens.Separate artificial toxin data for the three pathogens.Natural infection data for *Fusarium* spp. and *Aspergillus* spp.Natural toxin data for the three most important toxins.

It is suggested that different colors be used for the different risks. By this method, the similar or different responses of the hybrids can be identified easily. 

Generally, the toxin data are more important than the symptoms. Besides this, we have to consider the animal species to be fed. For dairy, low aflatoxin contamination is the most important; for ruminants, the DON and fumonisin levels can be higher. Swine is highly sensitive to DON, among others. 

The toxin contamination for a percentage of visual infection is variable. This is a significant new finding in the research. In other papers, this aspect was never analyzed. However, we can measure the toxin contamination for 1% of the ear rot and consider these results in presenting a more precise risk analysis [5,25]. The data show clearly that toxin data alone do not give enough support for solid decisions. Visual data are important per se to prove ear rot resistance differences, and when they are significant, we can evaluate the toxin data properly for toxin production intensity (toxin amount/one percent of ear infection, toxin overproduction, or relative toxin resistance, among others).

### 4.5. Resistance to Ear Rots, Toxin Contamination, and the Changing Climate

Considering climate change, mostly aflatoxins and fumonisins are mentioned [17,77,78]. Drought stress also induces increased AFB1 contamination [79]. In this respect, the two experiments provided convincing proof of very large variety differences in the same ecological scenario. In 2017/2018, the DON data (artificial inoculation) varied for the different hybrids between 10.84 and 63.27 mg/kg, the fumonisin data varied between 0.84 and 11.44 mg/kg, and the aflatoxin data varied between 19.7 and 338 μg/kg. The natural toxin contamination produced similar variations. In 2019/2020, at artificial inoculation, the DON varied between 7.48 and 101 mg/kg, the fumonisin varied between 0.87 and 6.66 mg/kg, and aflatoxin B1 varied between 53 and 1258 μg/kg. In the natural toxin contamination, the variation was similar, but in aAFB1, a variation between 0 and 1143 μg/kg was found. In different years and isolates, the data also varied, and it was important that several hybrids produced low variance, indicating stability. In others, the differences were extremely high, between 6 and 4947 μg/kg. Therefore, there is resistance to each fungal disease and their toxins. For this reason, this can be used to control mycotoxins, even in years supporting a high toxin contamination otherwise. We found similar data earlier, but this was the first test where aflatoxin and other Fusarium toxin responses could be checked in a single experiment [5,25,67,80].

We hoped that the introduction of ethanol production would solve the problem of contaminated grains. However, it made it even more severe. The toxins concentrated in the distilled dried grain with solubles (DGGS) by about 3–3.5-fold compared to the toxin contamination of the outgoing grain contamination [81,82]. Therefore, maize with more than 6 μg/kg of aflatoxin cannot be sold for ethanol production and will be given to animals. The higher resistance is also important in this respect because the food and feed safety required cannot otherwise be secured. 

### 4.6. Breeding Aspects

The information from these experiments is important for breeders. By identifying high yielding experimental hybrids with good or very good resistance to diseases and their toxins, they can look for common inbreds in such hybrids. This screening work helps to identify the proper inbred lines behind the hybrids, allowing for the breeding of more resistant hybrids. This allows one to initiate breeding programs at low cost and without screening hundreds of thousands of ears for resistance. 

An increased resistance level has other advantages. At present, without effective preharvest control knowledge, the toxin problem cannot be significantly mitigated. The Bt maize hybrids that are more resistance to ear rot will probably have a lower toxin contamination compared to the much more susceptible hybrids equipped with the same Bt gene(s). Similarly, a higher resistance to *A. flavus* and AFB1 would increase the efficacy of the biological control by the atoxic isolates of *A. flavus* in Africa and elsewhere. Therefore, resistance screening in local varieties of African maize and hybrids may also be significant. The increased genetic knowledge provides further possibilities to improve food and feed safety. 

## 5. Conclusions

Until now, maize food and feed safety was handled simply and ineffectively. It seems that the plant–pathogen system is very complex, with pathogens having different ecological and epidemic behaviors, often with plants showing differing toxin responses and a far more complicated genetic system than ever supposed before. However, it is possible to identify hybrids with qualities that allow food safety standards to be better met. This enables the possibility of screening various candidates and hybrids and exploiting this variability to increase food and feed safety. Considering the large differences between hybrids, breeding work can be more successful [83]. Hybrids were identified with good or very good resistance to different toxic species and their toxins. The selection of hybrids with disease and toxin resistance to toxigenic fungi is the first step. The next step is to start an effective breeding program that can further enhance multiresistant hybrids to the most important toxigenic fungi in the region in question. The resistance to toxigenic species is not immunity: this resistance should be supported with innovative agronomy, agronomic, and tilling practices, effective plant protection, and toxin control at harvest [14,84]. Manipulation of the grain and high-quality storage facilities are necessary to maintain or improve quality in the field at harvest and thereafter [85,86]. 

For abbreviations of toxin names, the EFSA opinions were considered [87,88].

## Figures and Tables

**Figure 1 jof-08-00293-f001:**
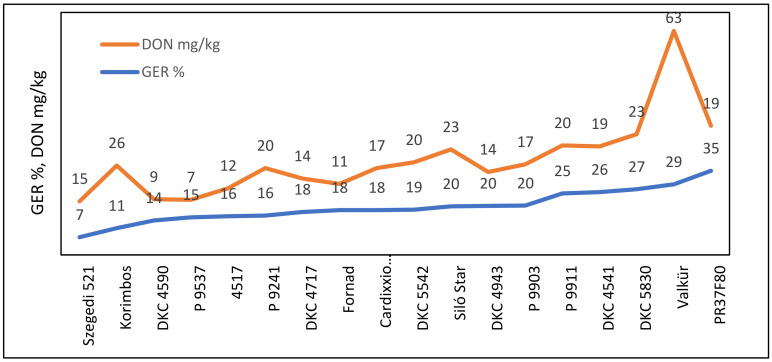
*F. graminearum* kernel ear rot percentage and DON contamination in 2017–2018. The LSD 5% value for DON is 15.52. For Valkür, the corresponding data point on the regression line is 27.70. Adding this to the LSD 5% value of 15.52 resulted in 43.21. Valkür had 63.28 mg/kg DON, 20.07 mg/kg more than the 43.21 mg/kg, so it is a DON overproducing hybrid. Variances for DON can be found for each data point in Appendix A.

**Figure 2 jof-08-00293-f002:**
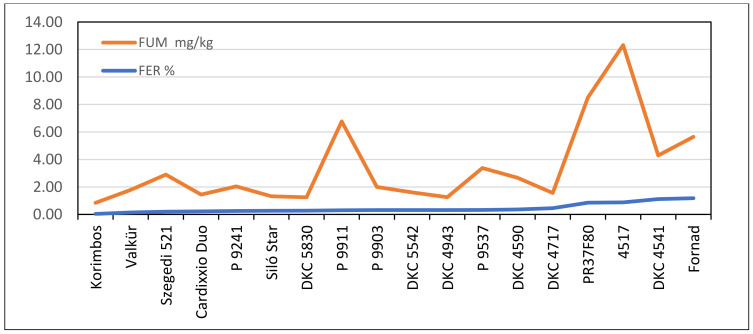
*F. verticillioides* kernel ear rot and fumonisin B_1_+B_2_ contamination in 2017–2018. The LSD 5% value for FB_1_+FB_2_ is 4.00. The distance between fumonisin’s corresponding point on the regression line and the actual FUM B_1_+B_2_ contamination for the P9911 and 4517 hybrids are 4.15 and 6.23, and they are larger than the LSD 5% value of 4.0. The fumonisin overproduction of these two hybrids is proven. Variances for the toxins can be found for each data point in Appendix A.

**Figure 3 jof-08-00293-f003:**
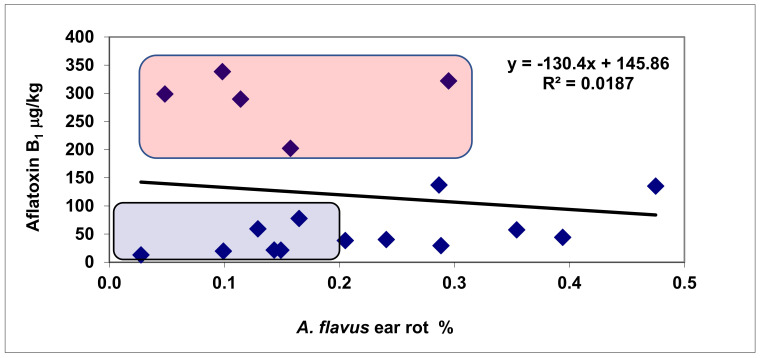
*A. flavus* ear rot and aflatoxin B_1_ in 2017–2018. The LSD 5% value for aflatoxin B1 is 66.29. Five hybrids show differences between data points and their corresponding points on the line higher than the LSD 5% value of 66.29 (between 76.9 and 214.6 mg/kg); they are considered toxin overproducers (rose color). Variances for the toxins can be found for each data point in Appendix A.

**Figure 4 jof-08-00293-f004:**
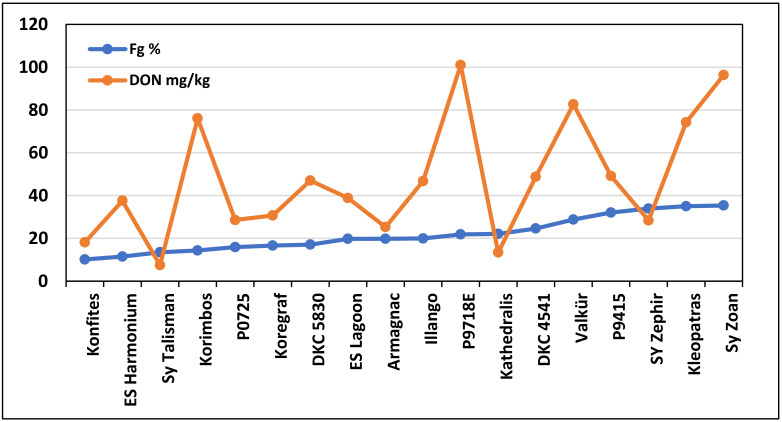
*F. graminearum* ear rot and DON production in the commercial hybrids following artificial inoculation (Szeged, 2019–2020). The variance can be found for each genotype in Appendix A.

**Figure 5 jof-08-00293-f005:**
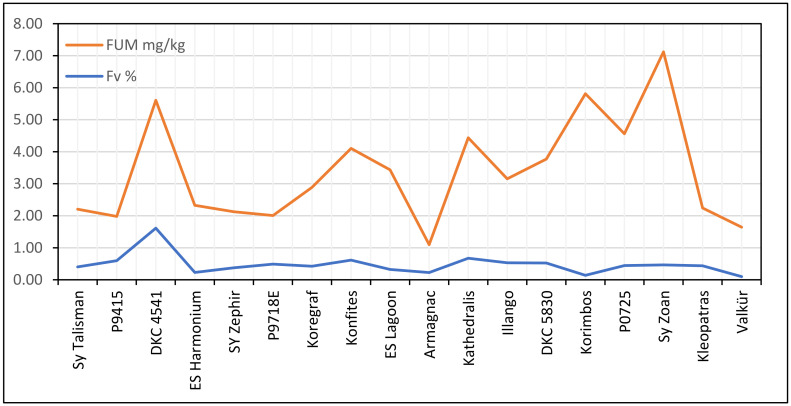
*F. verticillioides* ear rot and fumonisin B_1_+B_2_ production in the commercial hybrids following artificial inoculation (Szeged, 2019–2020). The variance for Fumonisin B_1_+B_2_ can be found for each genotype in Appendix A.

**Figure 6 jof-08-00293-f006:**
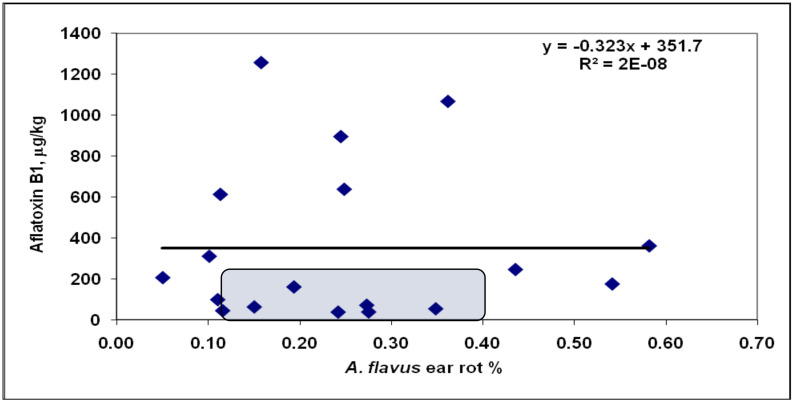
*A. flavus* ear rot and AFB1 production in the commercial hybrids following artificial inoculation (Szeged, 2019–2020). The four data sets differed so much that even large differences with a significant LSD could not be demonstrated. The variance for AFB1 can be found for each genotype in Appendix A.

**Figure 7 jof-08-00293-f007:**
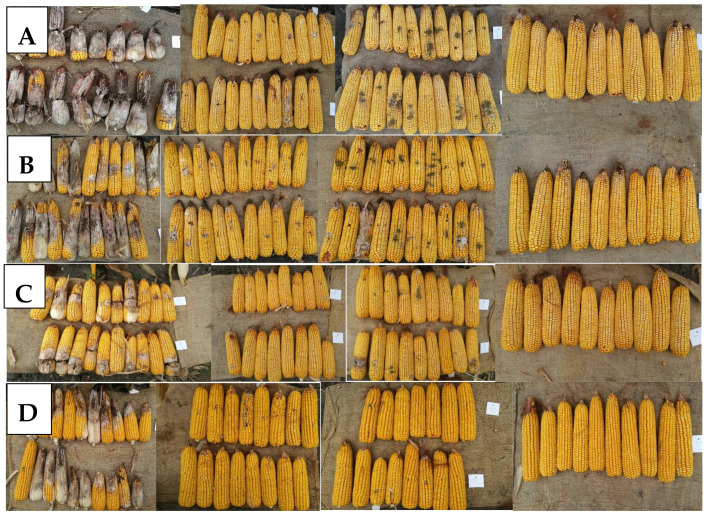
Visual symptoms of toxigenic fungi in 2019. *F. graminearum* is the first column on the left, followed by *F. verticillioides*, *A. flavus,* and the naturally infected control to the right. (**A**) P9415; (**B**) DKS 4541 (susceptible control); (**C**) Korimbos (resistant control); (**D**) Valkür. Columns 1–3: upper row: Isolate 1; lower row: Isolate 2; control contains one row.

**Figure 8 jof-08-00293-f008:**
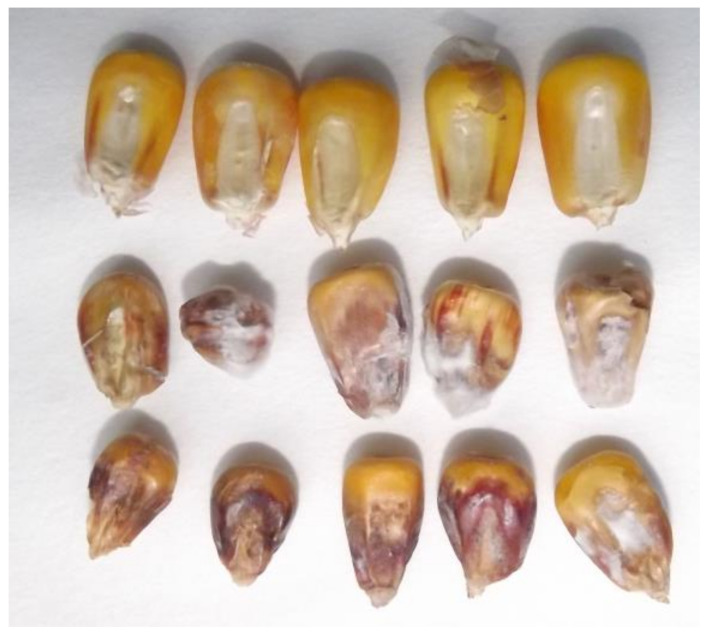
Maize grains from a naturally infected maize field. The upper row contains healthy grains without any visible sign of infection. The middle and lower lines contain grains whose upper part is mostly healthy, but the germ part is severely infected; in several grains, the whole surface is severely infected by *Fusarium* spp. (Courtesy: Mesterhazy).

**Figure 9 jof-08-00293-f009:**
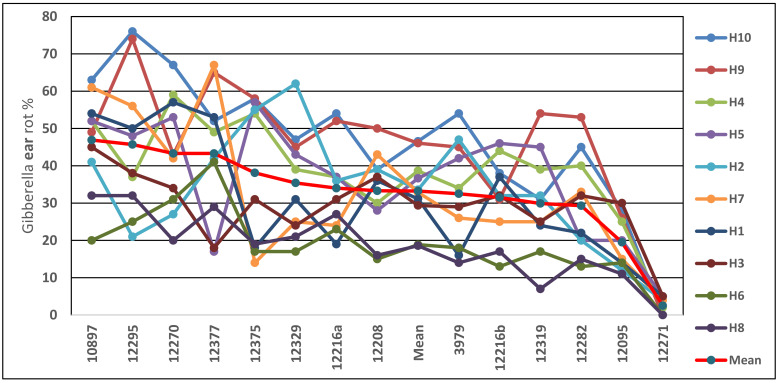
The GER kernel infection severity of 14 *F. graminearum* and *F. culmorum* isolates in maize hybrid resistance tests with 10 hybrids (H1-H10) [50].

**Table 1 jof-08-00293-t001:** Structure of a four-row plot. * Fg = *F. graminearum*, Fv = *F. verticillioides*, Af = *A. flavus*.

Parts of Row	Rows/Isolates			Check
2	Fg *. 2	Fv. 2.	Af. 2.	Check2
	way			
1	Fg. 1	Fv. 1	Af. 1.	Check1
	Row 1	Row 2	Row 3	Row 4

**Table 2 jof-08-00293-t002:** Optimized parameters of measured mycotoxins: deoxynivalenol (DON), aflatoxin B1 (AFB1), and fumonisin FB_1_+FB_2_). Lens parameters: collision cell exit potential (CXP), declustering potential (DP), and collision energy (CE).

Mycotoxin	Precursor Ion		Product Ions		Lens Parameters			Time Parameters	
	Q1 Mass (Da)	Adduct	Daughter	Q3 Mass (Da)	DP (V)	CE (V)	CXP (V)	Dwell (ms)	RT (min)
DON	355	[M+AcO]^−^	1	59.2	−40	−40	−8	150	4.32
		[M+AcO]^−^	2	295.2	−40	−16	−14	150	
AFB1	313	[M+H]^+^	1	285.0	176	35	14	20	5.47
		[M+H]^+^	2	241.0	176	53	14	20	
FB_1+_	722	[M+H]^+^	1	704.0	241	41	42	20	5.92
		[M]^+^	2	352.0	231	49	24	20	
FB_2_	706	[M+H]^+^	1	688.0	216	39	36	20	6.34
		[M]^+^	2	336.0	221	51	26	20	

**Table 3 jof-08-00293-t003:** Kernel resistance of commercial maize hybrids to toxigenic fungi following artificial inoculation and ear rot severity (%) (2017–2018).

Hybrid	Toxigenic Species			Check	Mean Ear Rot	Mean of Rankings	Variance in Rankings
	Fg	Fv	Af				
Szegedi 521	5.50	0.20	**0.11**	0.42	1.56	5.00	18.67
**Korimbos**	8.48	0.04	0.05	0.12	2.17	1.75	0.25
**DKC 4590**	11.18	0.36	0.15	0.50	3.05	9.50	25.67
**P9537**	11.64	0.32	0.14	0.25	3.09	7.25	11.58
**P9241**	12.21	0.25	0.10	0.23	3.20	5.00	0.67
4517	12.09	0.87	0.17	0.60	3.43	11.75	28.25
DKC 4717	13.29	0.45	0.24	0.35	3.58	10.50	9.67
Cardixxio Duo	13.88	0.23	0.29	0.26	3.66	8.25	17.58
**DKC 5542**	14.04	0.32	0.10	0.37	3.71	8.00	11.33
Siló Star	15.05	0.26	0.16	0.20	3.92	7.25	12.25
P9903	15.27	0.32	0.21	0.44	4.06	11.25	2.92
DKC 4943	15.22	0.32	0.39	0.32	4.06	12.00	14.00
Fornad	14.34	1.18	0.29	0.63	4.11	14.50	13.67
P9911	19.18	0.30	0.13	0.49	5.03	10.25	14.92
DKC 5830	20.38	0.27	0.30	0.21	5.29	10.50	35.00
DKC 4541	19.79	1.12	0.35	0.68	5.49	16.50	1.67
Valkür	21.83	0.14	0.03	0.03	5.51	5.25	61.58
PR37F80	26.20	0.86	0.48	0.58	7.03	16.50	3.00
Mean	14.98	0.43	0.20	0.37	4.00	9.50	15.70
LSD 5%	7.69	0.45	0.18	0.25	1.92		
Correlations	Fg	Fv	Af	Check	Mean	Ranks	
Fv	0.3190						
Af	0.5127 *	0.5746 *					
Check	0.1234	0.8271 ***	0.4918 *				
Mean	0.994 ***	0.4128	0.5691 *	0.2187			
Ranks	0.5810 *	0.8337 ***	0.8299 ***	0.7893 ***	0.6577 **		
Variance	0.2106	−0.1995	−0.2822	−0.3369	0.1702	−0.1790	
*** *p* = 0.001, ** *p* = 0.01, * *p* = 0.05		Fg = *F. graminearum*, Fv = *F. verticillioides*, Af = *. flavus*, **Bold names**: good general resistance					
Risk group		Low	Low to Medium		Medium to High		High

**Table 4 jof-08-00293-t004:** Summary table of the 2017–2018 kernel resistance tests of maize hybrids, ear rot severities, and toxin contamination at artificial and natural inoculation regimes.

Hybrid	Tox. Species, Ear Rot %			Toxins			Control					FAO
	Fg	Fv	Af	DON mg/kg	FB_1_+B_2_ mg/kg	AFB1 μg/kg	F. Ear Rot %	Af Ear Rot %	DON mg/kg	FB_1_+B_2_ mg/kg	AFB1 μg/kg	No.
Szegedi 521	5.50	0.20	0.11	14.80	2.70	289.75	0.42	0.01	0.15	0.65	90.50	560
Korimbos	8.48	0.04	0.05	25.8	0.81	298.75	0.12	0.00	2.40	0.42	27.50	575
**DKC 4590**	11.18	0.36	0.15	8.75	2.31	21.50	0.50	0.05	0.19	2.76	1.65	360
**P9537**	11.64	0.32	0.14	7.27	3.05	21.75	0.25	0.00	0.00	1.53	11.00	370
4517	12.09	0.87	0.17	11.58	11.44	77.75	0.60	0.00	0.00	7.05	32.00	520
**P9241**	12.21	0.25	0.10	19.57	1.79	19.75	0.23	0.00	0.00	0.71	14.00	350
DKC 4717	13.29	0.45	0.24	13.81	1.11	40.25	0.35	0.00	0.05	1.92	29.50	390
**Cardixxio Duo**	13.88	0.23	0.29	17.32	1.22	29.50	0.26	0.00	0.18	0.18	2.50	470
DKC 5542	14.04	0.32	0.10	19.54	1.29	338.50	0.37	0.00	0.00	2.71	33.00	540
Fornad	14.34	1.18	0.29	10.84	4.47	137.25	0.63	0.09	1.05	1.55	6.50	420
Siló Star	15.05	0.26	0.16	23.47	1.06	202.25	0.20	0.00	1.30	0.31	38.90	490
DKC 4943	15.22	0.32	0.39	13.98	0.94	44.25	0.32	0.05	0.32	0.43	21.50	400
P9903	15.27	0.32	0.21	17.04	1.69	38.50	0.44	0.00	0.10	0.41	2.50	390
P9911	19.18	0.30	0.13	19.79	6.47	59.25	0.49	0.17	0.00	1.60	10.50	450
DKC 4541	19.79	1.12	0.35	18.85	3.18	57.50	0.68	0.04	0.05	1.30	8.50	370
DKC 5830	20.38	0.27	0.30	22.72	0.98	322.00	0.21	0.00	0.00	0.09	2.50	560
Valkür	21.83	0.14	0.03	63.27	1.64	13.00	0.03	0.00	0.61	0.51	5.00	730
PR37F80	26.20	0.86	0.47	18.57	7.66	135.25	0.58	0.00	0.00	0.91	19.00	420
Mean	14.98	0.43	0.20	19.28	2.99	119.26	0.37	0.02	0.36	1.39	19.81	
LSD 5%	7.69	0.45	0.18	21.5	4.00	66.29	0.25					
Risk group		Low		Low to medium			Medium to high		High			

Fg = *F. graminearum*; Fv = *F. verticillioides*; Af = *A. flavus*; **highlights**: dark green = ow risk; light green = low–medium risk; yellow = medium–high risk; orange = high risk; **bold**: hybrids with good general resistance to all fungi and the control.

**Table 5 jof-08-00293-t005:** Kernel resistance test of commercial maize hybrids against toxigenic fungi and ear rot severity. Data are shown as percentages (2019–2020, Szeged, Hungary).

Hybrid	Toxigenic Species, Ear Rot %					Ranks	Ranks
	Fg^+^	Fv	Af	Check	Mean	Mean	Variance
Konfites	10.15	0.61	0.35	0.21	2.83	11.00	46.00
ES Harmonium	11.49	0.23	0.15	0.25	3.03	7.00	38.67
Sy Talisman	13.50	0.40	0.27	0.27	3.61	9.75	36.92
**Korimbos**	14.35	0.14	0.12	0.08	3.67	3.25	2.25
**P0725**	15.91	0.45	0.11	0.17	4.16	7.00	8.67
**Koregraf**	16.62	0.42	0.24	0.10	4.35	6.75	4.92
DKC 5830	17.08	0.52	0.44	0.17	4.55	11.00	18.00
**ES Lagoon**	19.76	0.32	0.24	0.19	5.13	8.75	8.92
**Armagnac**	19.79	0.23	0.11	0.13	5.06	5.00	8.00
Illango	19.94	0.53	0.36	0.18	5.25	12.25	6.92
P9718E	21.86	0.49	0.25	0.09	5.67	9.25	17.58
Kathedralis	22.10	0.67	0.58	0.21	5.89	15.25	7.58
DKC 4541	24.59	1.61	0.54	0.30	6.76	16.50	5.67
Valkür	28.78	0.10	0.05	0.05	7.25	4.25	42.25
P9415	32.04	0.60	0.28	0.24	8.29	14.50	1.00
SY Zephir	33.92	0.37	0.19	0.16	8.66	9.00	22.67
Kleopatras	35.02	0.44	0.10	0.16	8.93	8.75	38.92
Sy Zoan	35.36	0.47	0.16	0.18	9.04	11.75	20.92
Mean	21.79	0.48	0.25	0.17	5.67	9.50	18.66
LSD 5%	8.60	0.55	0.15	0.08	2.16	5.98	
Correlations	Fg ^+^	Fv	Af	Check	Mean	Ranks	
Fv	0.110						
Afl	−0.161	0.744 ***					
Check	−0.091	0.62 **	0.539 *				
Mean	0.998 ***	0.169	−0.108	−0.048			
Ranks, mean	0.254	0.799 ***	0.824 ***	0.704 ***	0.307		
Ranks, variance	−0.032	−0.257	−0.282	0.049	−0.046	−0.203	
*** *p* = 0.001, ** *p* = 0.01, * *p* = 0.05, Fg^+^ = *F. graminearum*; Fv = *F. verticillioides*; Af = *A. flavus.*							
Risk group:		Low	Low to medium		Medium to high		High

**Bold names**: low to middle low risk to all toxigenic species.

**Table 6 jof-08-00293-t006:** Kernel resistance tests of maize hybrids to toxigenic fungi. Summary of the second maize kernel resistance test and toxin analyses against three toxigenic fungi, 2019–2020.

Hybrid	Ear Rot % Art^+^.			Toxin Content Art.			Ear Rot, Control		Toxin Content, Control			
	Fg % ^x^	Fv %	Af %	DON mg/kg	FB_1_+B_2_ mg/kg	AFB1_1_ μg/kg	F. Check	Asp. Check	DON mg/kg	FB_1_+B_2_ mg/kg	AFB1 μg/kg	FAO No.
Konfites	10.15	0.61	0.35	18.15	3.49	53	0.21	0.000	1.70	1.88	2	430
**ES Harmonium**	11.49	0.23	0.15	37.60	2.10	62	0.25	0.000	0.13	0.81	4	380
**Sy Talisman**	13.50	0.40	0.27	7.48	1.80	71	0.27	0.000	0.72	0.54	2	250
Korimbos	14.35	0.14	0.12	76.08	5.67	44	0.08	0.000	3.49	0.30	408	575
P0725	15.91	0.45	0.11	28.58	4.12	613	0.17	0.005	0.00	0.12	794	560
**Koregraf**	16.62	0.42	0.24	30.72	2.47	37	0.10	0.000	0.00	0.30	352	410
DKC 5830	17.08	0.52	0.44	47.02	3.25	245	0.17	0.000	2.18	5.63	0	560
ES Lagoon	19.76	0.32	0.24	38.87	3.11	896	0.19	0.005	0.00	2.23	20	460
**Armagnac**	19.79	0.23	0.11	25.27	0.87	98	0.13	0.005	2.33	1.58	9	490
Illango	19.94	0.53	0.36	46.75	2.62	1068	0.18	0.008	0.05	0.75	1143	530
P9718E	21.86	0.49	0.25	101.01	1.52	638	0.09	0.000	0.17	1.17	3	390
Kathedralis	22.10	0.67	0.58	13.38	3.76	361	0.21	0.000	0.27	1.73	0	490
DKC 4541	24.59	1.61	0.54	48.77	4.00	174	0.30	0.005	3.59	3.06	0	370
Valkür	28.78	0.10	0.05	82.64	1.55	205	0.05	0.000	1.28	0.81	0	731
P9415	32.04	0.60	0.28	49.14	1.38	38	0.24	0.000	0.58	2.32	0	350
SY Zephir	33.92	0.37	0.19	28.38	1.75	160	0.16	0.005	0.63	2.36	4	390
Kleopatras	35.02	0.44	0.10	74.22	1.81	310	0.16	0.000	0.35	2.31	114	630
Sy Zoan	35.36	0.47	0.16	96.35	6.66	1258	0.18	0.008	0.00	1.65	0	560
Mean	21.79	0.48	0.25	47.24	2.88	352	0.17	0.002	0.97	1.64	159	
LSD 5%	8.60	0.55	0.15	54.90	3.789	528	0.08					
Risk group:	Low			Low to medium		Medium to high			High			

^x^ Fg = *F. graminearum*; Fv = *F. verticillioides*; Af = *A. flavus*; **bold hybrid names**: low and low-medium risk for all traits measured. Art^+^. = artificial inoculation.

**Table 7 jof-08-00293-t007:** Resistance and toxin relationships in commercial maize hybrid controls. Ear rot amount, toxin contamination, and the rate of toxin production for 1% ear rot severity (Szeged, 2017–2020). Ranking: *F. graminearum* kernel infection.

Hybrid	Fg/DON			Fv/FB_1_+B_2_			Af/AFB1		
	Fg %	mg/kg	DON/Fg %	Fv %	mg/kg	FB_1_+_2_/Fv %	Af %	mg/kg	AFB1/Af %
Korimbos	11.42	47.71	4.18	0.09	3.24	36.85	0.08	171.25	2091.60
DKC 5830	18.73	32.03	1.71	0.40	2.11	5.31	0.37	283.63	776.61
DKC 4541	22.19	31.47	1.42	1.37	3.59	2.63	0.45	115.88	258.82
Valkür	25.30	65.05	2.57	0.12	1.59	13.10	0.04	109.13	2816.13
Mean	13.08	44.06	3.37	0.49	2.63	5.34	0.22	169.97	759.81
LSD 5%	6.02	ns		0.59	ns		0.11	ns	

Fg = *F. graminearum*; Fv = *F. verticillioides*; Af = *A. flavus,* DON = deoxynivalenol; FB = fumonisin B_1_+B_2_; AFB1 = aflatoxin B_1_.

## Data Availability

All data are available upon reasonable request.

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
