# Peer review of "Updating the Methodology of Identifying Maize Hybrids Resistant to Ear Rot Pathogens and Their Toxins—Artificial Inoculation Tests for Kernel Resistance to Fusarium graminearum, F. verticillioides, and Aspergillus flavus"

_jof, 2022, doi:10.3390/jof8030293_

Round 1
Reviewer 1 Report
The risk evaluation of mycotoxins contamination in food and feed commodities is a very important point of the agronomical research: indeed, beside the relevance in terms of food safety for the final consumer, it is critial for the advancing in the development of containment strategies to be applied both in the field and during storage of grains and derivatives. Authors provided a complete and deepen study aimed at determining the definition of the correlation between the resistance to different mycotoxigenic species and the spread of their toxins in a recostruction experiment conducted in the field. A significan number of maize hybrids have been evaluated. I found the ms appropriatedly prepared and experiments well conducted.
I have some minor issues that must be addressed by Authors before I can consider the submission adequate for publication in JoF:
Abstract:
"Only about 10–15% of the hybrids showed higher resistance to all fungi and their toxins, a similar rate was susceptible to all, and the rest varied strongly." the sentence is scarcely clear and contains conceptual errors: to my knowledge, there are no scientific evidence that such mycotoxins are damaging for the plant health sensu strictu; when talking about fungi infection, Authors can use the words "resistance" and "susceptibility", but when referring to the toxin, they must avoid these terms. I suggest to change with "Only about 10–15% of the hybrids showed higher resistance to the fungal species tested and lower contamination level of their toxins". Please also change the relevant keyword and within thr whole main text.
"This means that resistance to different fungi......will result in a significant decrease in losses" this second part of the section is confunding and not clear at all. Are Authors reporting findings from their data of it's just a sort of general comment about the global issue of mycotoxins and phytopathogens spread in Hungary and in the rest of Europe? This must be completely re-written. Or, in alternative, the sentence could be moved above, leaving the experimental desing and results summary to be in the second part of the Abstract as it should be.
Introduction:
Due to the importance of toxigenic/atoxigenic A. flavus strains balance in the establishing of aflatoxin contamination on maize fields, I recommend to add a very little section dedicated to this topic, citing, for example, Authors that have investigated the problem of a correct identification of producing strains within the resident population, that is relevant to quantify the contamination risk (see "Facing the problem of "false positives": Re-assessment and improvement of a multiplex RT-PCR procedure for the diagnosis of A. flavus mycotoxin producers" International Journal of Food Microbiology. 129, 3, 300-305. DOI: 10.1016/j.ijfoodmicro.2008.12.016).
"three main toxigenic fungi: F. graminearum (deoxynivalenol and zearalenon), F. verticillioides (fumonisins), and A. flavus (aflatoxins)." the relevant toxin's name in paretheses should be introduced, and not simply put after the producing species.
"F. verticillioides; and A. flavus" delete punctuation
Authors show a pointed-list of their objectives: in my opinion this is not necessary, while They can simply list the aim of the work in a sentence. As an addition, objective "3. To determine how hybrids react to different species causing ear rot and how toxin productions depend on these species and their isolates;" is unclear: what are the "reactions" that will be evaluated? As an eco-physiological concept it must be reframed and explained...otherwise, it should be deleted.
Mat&Met:
"(Pioneer), Bayer (DeKalb), RAGT, Syngenta," all Companies information must be fully reported (city, state, an so on).
"The head of the consortium did not want to test the same hybrids across four years, as they wanted to have information about a possible high number of hybrids" this information is useless to the Reader.
Figures:
I strongly suggest Authors to move the Tables in Supplementary Materials, because they make the ms very hard to follow...
Author Response
Author's Reply to the Review Report (Reviewer 1)
Please provide a point-by-point response to the reviewer’s comments and either enter it in the box below or upload it as a Word/PDF file. Please write down "Please see the attachment." in the box if you only upload an attachment. An example can be found here.
Dear Reviewer 1,
Thank you for the evaluation of the paper. I followed your suggestions and I think the paper become better. I found several smaller mistakes, I improved them, and I found a better expression than it was in the previous version. You can follow all changes in the tracks. The comments to your suggestions are printed in red. I transferred further 5 tables int the Supplementary materials, but I would like to have your support to leave 6 tables in the paper as listed in the last paragraph.
Yours very sincerely
Akos Mesterhazy
* Author's Notes to Reviewer
FileEditViewInsertFormatToolsTableHelp
Paragraph
P
0 WORDS
Word / PDF
or
Review Report Form
Open Review
(x) I would not like to sign my review report
( ) I would like to sign my review report
English language and style
( ) Extensive editing of English language and style required
( ) Moderate English changes required
(x) English language and style are fine/minor spell check required
( ) I don't feel qualified to judge about the English language and style
Yes Can be improved Must be improved Not applicable
Does the introduction provide sufficient background and include all relevant references? (x) ( ) ( ) ( )
Is the research design appropriate? (x) ( ) ( ) ( )
Are the methods adequately described? ( ) (x) ( ) ( )
Are the results clearly presented? ( ) (x) ( ) ( )
Are the conclusions supported by the results? (x) ( ) ( ) ( )
Comments and Suggestions for Authors
The risk evaluation of mycotoxins contamination in food and feed commodities is a very important point of the agronomical research: indeed, beside the relevance in terms of food safety for the final consumer, it is critial for the advancing in the development of containment strategies to be applied both in the field and during storage of grains and derivatives. Authors provided a complete and deepen study aimed at determining the definition of the correlation between the resistance to different mycotoxigenic species and the spread of their toxins in a recostruction experiment conducted in the field. A significan number of maize hybrids have been evaluated. I found the ms appropriatedly prepared and experiments well conducted.
I have some minor issues that must be addressed by Authors before I can consider the submission adequate for publication in JoF:
Abstract:
"Only about 10–15% of the hybrids showed higher resistance to all fungi and their toxins, a similar rate was susceptible to all, and the rest varied strongly." the sentence is scarcely clear and contains conceptual errors: to my knowledge, there are no scientific evidence that such mycotoxins are damaging for the plant health sensu strictu; when talking about fungi infection, Authors can use the words "resistance" and "susceptibility", but when referring to the toxin, they must avoid these terms. I suggest to change with "Only about 10–15% of the hybrids showed higher resistance to the fungal species tested and lower contamination level of their toxins". Please also change the relevant keyword and within thr whole main text.
I changed the text to the you suggested. I did not intend to speak about the toxin damage to plants. We did not check it, and there is one paper Lemmens et al. 2005 that looks for resistance to DON in wheat, but nothing this type is for corn. The only thing we wanted to make clear that the toxin contamination can be governed by other genetic factors than resistance to disease. I think that your suggestion provides a good solution and excludes misinterpretation. Thank you. The resistance to toxin is a common expression in the literature, the authors think that the two resistances are synonymous expressions. So, it is not an accident that we used resistance to toxin expression. We know now that this is not generally true, but you are right, the content of the expression should be defined correctly and not use as a scientific slang.
"This means that resistance to different fungi......will result in a significant decrease in losses" this second part of the section is confunding and not clear at all. Are Authors reporting findings from their data of it's just a sort of general comment about the global issue of mycotoxins and phytopathogen s spread in Hungary and in the rest of Europe?
We can conclude only for Hungary as we have data only from this country. I think, every country should check its situation of aflatoxin contamination and make decisions when the natural contamination is higher in several hybrids than the officially allowed limit value. This is expressed with the remarks that other countries may be jeopardized in the region. This sentence was inserted higher in the texts from its present position.
This must be completely re-written. Or, in alternative, the sentence could be moved above, leaving the experimental design and results summary to be in the second part of the Abstract as it should be.
I made it, I hope it will be clearer.
Introduction:
Due to the importance of toxigenic/atoxigenic A. flavus strains balance in the establishing of aflatoxin contamination on maize fields, I recommend to add a very little section dedicated to this topic, citing, for example, Authors that have investigated the problem of a correct identification of producing strains within the resident population, that is relevant to quantify the contamination risk (see "Facing the problem of "false positives": Re-assessment and improvement of a multiplex RT-PCR procedure for the diagnosis of A. flavus mycotoxin producers" International Journal of Food Microbiology. 129, 3, 300-305. DOI: 10.1016/j.ijfoodmicro.2008.12.016).
I made a short new paragraph to mention the problem and used the reference you suggested. Thank you for the idea. It can explain why theoretically toxin producers with false positivity cannot produce aflatoxin even the fungus could do it. This explains our negative aflatoxin result in the paper from 2018. Now we can check our isolates for this trait. The case of nontoxic isolates with lacking aflatoxin cluster is clear needs no comment. In order not to increase the number of references I changed three for the new shoer paragraph, highlighted by yellow in the references.
"three main toxigenic fungi: F. graminearum (deoxynivalenol and zearalenon), F. verticillioides (fumonisins), and A. flavus (aflatoxins)." the relevant toxin's name in paretheses should be introduced, and not simply put after the producing species.
"F. verticillioides; and A. flavus" delete punctuation
I rephrased the sentence accordingly.
Authors show a pointed-list of their objectives: in my opinion this is not necessary, while They can simply list the aim of the work in a sentence. As an addition, objective "3. To determine how hybrids react to different species causing ear rot and how toxin productions depend on these species and their isolates;" is unclear: what are the "reactions" that will be evaluated? As an eco-physiological concept it must be reframed and explained...otherwise, it should be deleted.
I pooled them into one paragraph, I omitted point three, the other objectives contain it.
Mat&Met:
"(Pioneer), Bayer (DeKalb), RAGT, Syngenta," all Companies information must be fully reported (city, state, an so on).
I added the firms and addresses of their Hungarian firms.
"The head of the consortium did not want to test the same hybrids across four years, as they wanted to have information about a possible high number of hybrids" this information is useless to the Reader.
I omitted it.
Figures:
I strongly suggest Authors to move the Tables in Supplementary Materials, because they make the ms very hard to follow...
I carefully screened the tables which could be moved to the Supplementary Materials. As this paper is not only for basic research that would seek new scientific ideas and results, but it has also a plant breeding connection, therefore I think that the main text should contain the most important tables where the readers can see the hybrid differences. I know breeders and how they think. I do not think that these remaining tables would disturb them in understanding the message. When they need more data for understanding, they can look the supplementary tables. In the Materials and Methods, the toxin table 1 is important when somebody would like to repeat the test, could do it with the same methodology which was used here. From Experiment 1 I left only Table 2 for the resistance data and Table 3 for the summary. From the 2nd test this is the same, and we need the Table 6 to demonstrate the performance of the control hybrids in the four year’s testing. Now we have 9 tables in the supplementary material.
Submission Date
29 January 2022
Date of this review
04 Feb 2022 11:11:40
Anwer: 09 March 2022 9:30
Reviewer 2 Report
The current study shows the difference between maize hybrid resistance to three mycotoxin producing pathogenic fungi namely Fusarium graminearum, Fusarium verticillioides and Aspergillus flavus. This study is novel and relevant to farmers in the sense that it focuses on hybrid instead of inbred response to the disease and accumulated mycotoxin. I support its publication in the present state.
Author Response
Dear Reviewer 2
Thank you for your work to improve this paper and the support I received from you during the editorial process.
Yours very sincerely
Akos Mesterhazy